

# KdV charges in $T\bar{T}$ theories and new models with super-Hagedorn behavior

Bruno Le Floch[1*] and Márk Mezei[2]

**1** Philippe Meyer Institute, Physics Department,
École Normale Supérieure, PSL Research University, Paris, France
**2** Simons Center for Geometry and Physics, SUNY, Stony Brook, USA

* bruno.le.floch@ens.fr

## Abstract

Two-dimensional CFTs and integrable models have an infinite set of conserved KdV higher spin currents. These currents can be argued to remain conserved under the $T\bar{T}$ deformation and its generalizations. We determine the flow equations the KdV charges obey under the $T\bar{T}$ deformation: they behave as probes "riding the Burgers flow" of the energy eigenvalues. We also study a Lorentz-breaking $T_{s+1}\bar{T}$ deformation built from a KdV current and the stress tensor, and find a super-Hagedorn growth of the density of states.

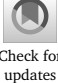
# 1  Introduction and summary

The $T\bar{T}$ deformation of two-dimensional field theories has attracted significant attention recently due to its connection to disparate directions of research. It is a universal (and often leading) irrelevant operator near the infrared fixed point of renormalization group flows [1–3]. The $T\bar{T}$ deformation greatly increases the space of known integrable theories [4–6]. A novel deformation of S-matrices [7–9] was understood to be equivalent to the $T\bar{T}$ deformation of the Lagrangian [10–12], and also led to an alternative description as matter coupled to flat space Jackiw-Teitelboim gravity. See also [13]. Its relationship to the holographic renormalization group was explored in [14–23]. $T\bar{T}$-deformed theories and their generalizations share features with little string theories that are holographically dual to asymptotically linear dilaton backgrounds. This connection was explored in [24–34].

    Partition functions in $T\bar{T}$-deformed theories have been computed with a multitude of methods. The torus partition function was determined by a path integral over random metrics in [11,35] and it has been proven to be a unique modular covariant partition function satisfying certain conditions [36–38]. The $S^2$ partition function was computed using large-$N$ factorization in [39,40] and in the $T\bar{T}$-deformed two-dimensional Yang-Mills theory in [41], see also [42] for analysis of the theory put on $S^2$. Entanglement entropies were computed using the replica trick in [39,43–50].

    Other solvable irrelevant deformations were considered in [51–60]. Closed form Lagrangians often provide important insight into these deformed theories, and many have been

constructed in [5,57,58,61–64]. Correlation functions were investigated in [2,16,19,27,55,65]. The interplay between the $T\bar{T}$ deformation and supersymmetry was explored in [64,66–69]. The S-matrix of various worldsheet theories has been connected to the $T\bar{T}$ deformation in [70–73]. For a pedagoical introduction, see [74].

In this paper, we continue the quest of finding solvable examples of spectra of quantum field theories deformed by irrelevant operators. The first such example was provided by the pioneering papers [4,5] for $T\bar{T}$-deformed theories and a very simple extension was solved in [22]. The spectrum of the $J\bar{T}$-deformed CFTs was obtained in [28], completing the work of [51]. In [57], we used background fields to determine the spectrum of CFTs deformed by irrelevant operators built from $J_\mu$, $\bar{J}_\mu$, $T_{\mu\nu}$, where the former are the (anti)holomorphic $U(1)$ currents of the theory and the latter is the stress tensor. Some steps in the derivation of [57] (and also in the determination of the $J\bar{T}$-deformed spectrum in [28]) were conjectural and only backed up by various checks. In contrast, in this paper we derive rigorously the flow of the quantum KdV charges [75] under the $T\bar{T}$ deformation, and determine the energy spectrum, KdV charges, and asymptotic density of states in the zero momentum sector under a $T_{s+1}\bar{T}$ deformation starting from a CFT. We often refer henceforth to the theory which we start deforming as the seed theory.

It was shown in [4,5] that the energy spectrum of $T\bar{T}$-deformed relativistic theories on the cylinder is governed by the equation

$$\partial_\lambda E_n = -\pi^2 \left( E_n \partial_L E_n + \frac{P_n^2}{L} \right), \tag{1.1}$$

where $E_n$ and $P_n$ are the energy and momentum eigenvalues, $L$ is the circumference of the circle, and $\lambda$ is the deformation parameter. In this paper we derive that the quantum KdV charges $\langle P_s \rangle_n$ of the eigenstate $|n\rangle$, if present in the seed theory, obey

$$\partial_\lambda \langle P_s \rangle_n = -\pi^2 \left( E_n \partial_L \langle P_s \rangle_n + P_n \frac{s \langle P_s \rangle_n}{L} \right). \tag{1.2}$$

The allowed values of $s$ are $\pm 1$, $\pm 3$, .... This equation was also obtained using integrability techniques of [5,58]. Our field theory derivation applies more broadly, to the $T\bar{T}$ deformation of any Lorentz-invariant theory that contains at least one higher spin conserved charge, and hence rules out the possibility that the evolution equation (1.2) is a miracle of some special models.

There is a beautiful analogy with hydrodynamics. Equation (1.1) is the forced inviscid Burgers equation

$$\partial_t u + u \, \partial_x u = -\frac{p^2}{x^3}, \tag{1.3}$$

where the right-hand side is the forcing term, and we made the identifications

$$u \equiv E_n, \quad t \equiv \pi^2 \lambda, \quad x \equiv L, \tag{1.4}$$

and used that $P_n = p/L$. Then (1.2) is translated to

$$\partial_t P_s + u \, \partial_x P_s = -\frac{sp}{x^2} P_s, \tag{1.5}$$

which has the interpretation of particles probing the Burgers flow (but not backreacting on it): the left hand side is the material derivative of $P_s$ and the right hand side is a forcing term. This equation is referred to as a passive scalar equation in the fluid dynamics literature, see the elegant review [76]. Admittedly, we have not encountered the particular forcing term in (1.5) in the fluid dynamics literature. For a CFT seed theory, we also solve these equations.

As the second major result of the paper, we obtain the evolution of the spectrum for $T_{u+1}\bar{T}$-deformed relativistic theories, where $T_{u+1}$ is the current of the KdV charge $P_u$, in the zero momentum sector:

$$\partial_\lambda \langle P_s \rangle_n = 2\pi^2 \langle P_u \rangle_n \partial_L \langle P_s \rangle_n \qquad \text{if } P_n = 0 \,. \tag{1.6}$$

We are not able to derive a closed set of equations for sectors with $P_n \neq 0$ that would generalize the equation above. Solving these equations for a CFT seed theory, we find that the eigenstates that start their lives as primaries in the CFT exhibit super-Hagedorn asymptotic density of states

$$\rho_{\text{primary}}(E) \approx \exp\left( \sqrt{\#(c-1)\lambda}\, E^{(|u|+1)/2} \right), \tag{1.7}$$

where $\#$ is a number that we determine and $c$ is the central charge of the seed CFT.

Let us indicate the major steps in our derivation of the two main results by giving the outline of the paper. Section 2 is largely a review of [4]. We introduce the higher spin KdV currents and their charges, operators that have factorizing expectation values, and show that deforming a theory by quadratic composites of KdV currents preserve these symmetries. New results presented in this section are: the proof of factorization without the non-degeneracy assumption on the energy spectrum (which is important for CFTs, where there are many states degenerate in energy in a Virasoro module); the proof that the (possibly non-abelian) algebra of charges does not get deformed, and hence the KdV charges continue to commute in the deformed theory proving a conjecture made in [4]; and the generalization of the factorization property to new composite operators that are products of arbitrarily many factors. In Section 3 we use these results to derive an evolution equation for the KdV charges. An important step in the derivation is a novel formula for the expectation value of the space component of a KdV current as a length-derivative of the KdV charge in Lorentz-invariant theories. In Section 4 we apply results of Section 2 to a $T_{u+1}\bar{T}$ deformation. Superficially similar deformations were analyzed in [58] and our results partially agree despite fundamental differences in the two deformations, which we explain in Appendix F. Other appendices discuss various technical points used in the main text.

## 2 Change of KdV currents under irrelevant deformations

An important property of the class of irrelevant deformations built from an antisymmetric product of currents considered in [4] is that they preserve many symmetries of the undeformed theory: any current whose charge commutes with the charges of the currents building the deformation can be adjusted so that it remains conserved in the new theory. In the case of $T\bar{T}$ these are the currents that do not involve the coordinates explicitly. See Appendix G for a derivation of these facts.

### 2.1 KdV currents and the $A^s_\sigma$ operators

Let us consider the $T\bar{T}$ deformation of a CFT first. Since the dilation current of a CFT, $j^{(D)}_\mu = T_{\mu\nu} x^\nu$ depends on the coordinates explicitly, dilation is not a symmetry of the deformed theory. Similarly the currents whose charges are the Virasoro generators $L_n$ with $n \neq 0$ cannot be adjusted to remain conserved, thus most of the conformal group is lost. There is still a remnant of the infinite symmetry algebra in the deformed theory, and the maximal commuting

set is formed by the KdV currents and charges, which in a CFT take the form

$$T_{s+1} = :T^{\frac{s+1}{2}}: + \ldots$$
$$P_s = \frac{1}{2\pi} \int dz \; T_{s+1}(z),$$

(2.1)

where the $\ldots$ stand for terms that involve derivatives and lower powers of the stress tensor. These can be adjusted to remain conserved after deformation, namely

$$0 = \bar{\partial} \, T_{s+1} - \partial \, \Theta_{s-1}$$
$$P_s = \frac{1}{2\pi} \int \left( dz \; T_{s+1} + d\bar{z} \; \Theta_{s-1} \right).$$

(2.2)

To show that this is indeed possible we have to review the methods of [4]. We work in the Hamiltonian formalism.

In a CFT the algebra of conserved charges is the universal enveloping algebra of the Virasoro algebra $\mathcal{U}(\mathrm{Vir})$, which is formed by the sums of products of $L_n$'s.[1] In fact, a noncommutative subalgebra of charges generated by $\prod_{n_i} L_{n_i}$, where $\sum n_i = 0$ is also preserved (with undeformed structure constants) by the $T\bar{T}$ deformation. We develop this direction in Appendix G. In the main text, we focus on the KdV charges only. These charges are also preserved in integrable massive deformations of minimal models.

To avoid needlessly duplicating later equations, we denote by $P_{-s}$, $T_{-s+1}$, $\Theta_{-s-1}$ the charges and currents denoted by $\bar{P}_s$, $\bar{\Theta}_{s-1}$, $\bar{T}_{s+1}$ in [4]. For $s = 1$, we get the left- and right-moving combinations of $H$ and $P$ that act as derivatives on local operators with no explicit coordinate dependence:

$$P_{\pm 1} = -\frac{H \pm P}{2},$$
$$[P_1, \mathcal{O}] = -i\partial \mathcal{O}, \qquad [P_{-1}, \mathcal{O}] = i\bar{\partial}\mathcal{O}.$$

(2.3)

Many derivations below are simplified by using these commutators instead of derivatives. Next, we use the fact that the commutativity of charges $[P_s, P_\sigma] = 0$ is equivalent to the integral of $[P_s, dz \; T_{s+1} + d\bar{z} \; \Theta_{s-1}]$ vanishing on any cycle, hence it is an exact one-form [4], which in commutator language can be written as:[2]

$$[P_\sigma, T_{s+1}(z, \bar{z})] = [P_1, A_\sigma^s(z, \bar{z})],$$
$$[P_\sigma, \Theta_{s-1}(z, \bar{z})] = -[P_{-1}, A_\sigma^s(z, \bar{z})],$$

(2.4)

i.e. the commutators on the LHS give derivatives of a local operator $A_\sigma^s$ that is only defined up to addition of the identity. The unnatural position of indices in $A_\sigma^s$ simplifies notations for antisymmetrization later on. From the definitions it follows that

$$A_1^s = T_{s+1}, \qquad A_{-1}^s = -\Theta_{s-1}.$$

(2.5)

In Appendix A.1 we analyze some further basic identities obeyed by $A_\sigma^s$. In Appendix B we show in Lorentz invariant theories that

$$A_s^1 = s \, T_{s+1}, \qquad A_s^{-1} = s \, \Theta_{s-1}.$$

(2.6)

---

[1]There is an antiholomorphic copy as well. These charges are integrals of local holomorphic currents such as $z^n T^m$. These charges in general do not commute with the Hamiltonian $H$, they are conserved because their noncommutativity with $H$ is compensated by their explicit time dependence. The maximal commuting set of these charges are the KdV charges.

[2]In the notations of [4] our operators $A_\sigma^s$ are equal to $iA_{\sigma,s}$ for $0 < \sigma, s$, $iB_{\sigma,-s}$ for $s < 0 < \sigma$, $i\bar{A}_{-\sigma,-s}$ for $\sigma, s < 0$, and $i\bar{B}_{-\sigma,s}$ for $\sigma < 0 < s$.

## 2.2 Factorizing operators

Let us introduce the bilinear operators of [4]:

$$X^{st}(z) \equiv \lim_{x \to z} \left( T_{s+1}(x)\Theta_{t-1}(z) - \Theta_{s-1}(x)T_{t+1}(z) \right) + \text{(reg. terms)}, \qquad (2.7)$$

which are spin-$(s+t)$ operators. The regulator terms are total derivatives and do not play an important role in any of the subsequent results. As we show in Appendix C improvement transformations of the currents only change the regulator terms, and hence drop out from subsequent results. A special case is the bilinear operator

$$X^{s,-s}(z) \equiv \lim_{x \to z} \left( T_{s+1}(x)\bar{T}_{s+1}(z) - \Theta_{s-1}(x)\bar{\Theta}_{s-1}(z) \right) + \text{(reg. terms)}, \qquad (2.8)$$

that we can informally call "$T_{s+1}\bar{T}_{s+1}$" following the precedent set by the usage of $T\bar{T}$ in the literature. These composite operators defined by point splitting have remarkable properties. They obey factorization, in joint eigenstates of all KdV charges on the cylinder $S^1 \times \mathbb{R}$,

$$\langle n|X^{st}|n\rangle = \langle n|T_{s+1}|n\rangle\langle n|\Theta_{t-1}|n\rangle - \langle n|\Theta_{s-1}|n\rangle\langle n|T_{t+1}|n\rangle, \qquad (2.9)$$

where we omitted writing the arguments of operators, as one point functions in eigenstates do not depend on the position of the operator. In Appendix A.3 we present an algebraic proof of (2.9), which relaxes the assumption of non-degenerate energy spectrum that was needed in [4], and also allows for the generalization of factorization to the operator:

$$\begin{aligned} \mathcal{X}^{s_1 \ldots s_k}_{\sigma_1 \ldots \sigma_k} &\equiv k! \left( A^{s_1}_{[\sigma_1} \cdots A^{s_k}_{\sigma_k]} \right)_{\text{reg}}, \\ \langle n|\mathcal{X}^{s_1 \ldots s_k}_{\sigma_1 \ldots \sigma_k}|n\rangle &= k! \langle n|A^{s_1}_{[\sigma_1}|n\rangle\langle n|A^{s_2}_{\sigma_2}|n\rangle \cdots \langle n|A^{s_k}_{\sigma_k]}|n\rangle. \end{aligned} \qquad (2.10)$$

Note that $\mathcal{X}^{st}_{-1,1} = X^{st}$ defined in (2.7). The point-splitting regularization $(\bullet)_{\text{reg}}$ is detailed in Appendix A.2.

## 2.3 Special deformations preserve the KdV charges

As was stated at the beginning of this section, deforming the theory by $X^{st}$ preserves the symmetries (2.2). The proof proceeds by constructing the deformation of conserved currents under the irrelevant deformation. Let us assume that we deform the Hamiltonian by $\int dy\, X(y)$ and ask what conditions need to be satisfied so that the current with charge $P_s$ remains conserved.

First we linearize $[H, P_s] = 0$ in the coupling of $X$ to obtain:

$$0 = [\delta H, P_s] + [H, \delta P_s] = \int dy\, [X(y), P_s] + [H, \delta P_s]. \qquad (2.11)$$

This equation can only be satisfied if $[P_s, X(y)]$ is a total derivative, i.e. there exist $Y_{\pm 1}$ such that

$$\boxed{[P_s, X(y)] = [P_{-1}, Y_{-1}(y)] + [P_1, Y_1(y)],} \qquad (2.12)$$

and then

$$\delta P_s = -\frac{1}{2} \int dy\, (Y_{-1}(y) + Y_1(y)) \qquad (2.13)$$

obeys (2.11) thanks to the fact that the integral of a derivative vanishes, which we use in the form $\int dy\, [P, Y_1(y) - Y_{-1}(y)] = 0$. In Appendix D we show that the condition (2.12) (with $Y_{\pm 1}$ local) is enough to ensure that $P_s$ remains the integral of a local conserved current.

We just saw that the currents remain conserved if $X$ satisfies the condition (2.12). Let us check that (2.12) is obeyed for $X = X^{tu}$. We remind ourselves that according to (2.10)

$$X^{tu}(y) = 2 \lim_{x \to y} A^t_{[-1}(x) A^u_{1]}(y) + \text{(reg. terms)}. \tag{2.14}$$

For details on the regulator terms see Appendix A.2. Using (A.6) we find

$$[P_s, X^{tu}(y)] = [P_{-1}, \mathcal{X}^{tu}_{s,1}] + [P_1, \mathcal{X}^{tu}_{-1,s}], \tag{2.15}$$

where we used the definition (2.10). From (2.15) we conclude that (2.12) is obeyed with $Y_{-1} = \mathcal{X}^{tu}_{s,1}$ and $Y_1 = \mathcal{X}^{tu}_{-1,s}$.

In summary, we have that under $X^{tu}$ deformation:

$$\delta P_s = -\frac{1}{2} \int dy \, \left( \mathcal{X}^{tu}_{s,1}(y) + \mathcal{X}^{tu}_{-1,s}(y) \right). \tag{2.16}$$

Because we can add $T_{\sigma+1} + \Theta_{\sigma-1}$, the time component of a conserved current (with commuting charge) to the integrand in (2.16), there is some ambiguity in (2.16). Ambiguities are discussed and partially resolved in Appendix C. It can be verified that (2.16) leads to $\delta P = 0$, $\delta H = \int dy \, X^{tu}(y)$ as assumed. (If it did not, we would have had to shift $Y_{\pm1}$ found in (2.15) by some conserved current to make the story consistent.)

A generalization of (2.15), namely (A.14), states in particular that $[P_s, \mathcal{X}^{tu}_{r,\pm1}] - [P_r, \mathcal{X}^{tu}_{s,\pm1}] = [P_{\pm1}, \mathcal{X}^{tu}_{rs}]$, which implies

$$\begin{aligned}\delta[P_r, P_s] &= -[P_s, \delta P_r] + [P_r, \delta P_s] = \frac{1}{2} \int dy \, \left( [P_s, \mathcal{X}^{tu}_{r,1}(y) + \mathcal{X}^{tu}_{-1,r}(y)] - r \leftrightarrow s \right) \\ &= \frac{1}{2} \int dy \, [P_1 - P_{-1}, \mathcal{X}^{tu}_{rs}(y)] = 0,\end{aligned} \tag{2.17}$$

where we used that $[P_1 - P_{-1}, \mathcal{O}] = -i\partial_x \mathcal{O}$ integrates to zero. This proves Smirnov and Zamolodchikov's conjecture in [4] that the $T\bar{T}$ deformation leaves the (adjusted) KdV charges commuting. We explain in Appendix G which parts of the story presented here generalize to nonabelian charges that do not commute with the KdV charges and to internal symmetry charges.

We remark that while the operators $\mathcal{X}^{s_1...s_k}_{\sigma_1...\sigma_k}$ defined in (2.10) retain many of the nice properties of $X^{tu}$, deforming by them does not preserve the KdV charges, as the condition (2.12) is not satisfied for them. There is one other, somewhat trivial deformation that preserves the KdV charges, the deformation by $A^t_1 = T_{t+1}$ or $A^t_{-1} = -\Theta_{t-1}$. Note that these operators can also be written as $\mathcal{X}^t_{\pm1}$. These deformations by conserved current components correspond to turning on background gauge fields. The condition (2.12) is satisfied with $Y_{-1} = 0$, $Y_1 = A^t_s$ (for the $A^t_1$ deformation) and $Y_{-1} = A^t_s$, $Y_1 = 0$ (for the $A^t_{-1}$ deformation), which is verified from the definitions (2.4). This choice is not good enough however, as it leads to $\delta P = \pi P_t$. This problem can be taken care of by using the ambiguity discussed below (2.16) of adding conserved currents to $Y_{\pm1}$. We work out the example of $A^t_1$, as the $A^t_{-1}$ case can be treated in complete analogy. We shift

$$\begin{aligned}\text{for } s = 1: \quad & Y_{-1} = 0, \quad && Y_1 = A^t_1 = T_{t+1}, \\ \text{for } s = -1: \quad & Y_{-1} = T_{t+1}, \quad && Y_1 = A^t_{-1} + \Theta_{t-1} = 0,\end{aligned} \tag{2.18}$$

which then gives $\delta P = 0$, $\delta H = \int dy \, A^t_1$ as required.

## 3 Evolution of the spectrum of KdV charges

### 3.1 Evolution under generic deformations

In Section 2 we understood how the KdV charges change under irrelevant deformations. Let us now choose joint eigenstates $|n\rangle$ of the commuting charges $P_s$, and denote their eigenvalues by $\langle P_s \rangle_n \equiv \langle n|P_s|n\rangle$. We can use the Hellman-Feynman theorem for the infinitesimal deformation $\delta \langle P_s \rangle_n = \langle n|\delta P_s|n\rangle$ to write:

$$
\begin{aligned}
\partial_\lambda \langle P_s \rangle_n &= -\frac{L}{2} \langle \mathcal{X}^{tu}_{s,1}(y) + \mathcal{X}^{tu}_{-1,s}(y) \rangle_n \\
&= \frac{L}{2} \Big( \langle A^t_1 - A^t_{-1} \rangle_n \langle A^u_s \rangle_n - \langle A^t_s \rangle_n \langle A^u_1 - A^u_{-1} \rangle_n \Big) \\
&= \pi \Big( \langle P_t \rangle_n \langle A^u_s \rangle_n - \langle P_u \rangle_n \langle A^t_s \rangle_n \Big),
\end{aligned}
\tag{3.1}
$$

where we introduced $\lambda$ as the coupling constant of $X^{tu}$, in the first line we used (2.16) and the spacetime independence of one point functions in energy eigenstates to evaluate the space integral, in the second line we used factorization (2.10), and in the third we used (2.5). We obtained an evolution equation for the change in the spectrum of conserved charges under irrelevant deformations.[3] That such an equation can be derived is already remarkable, but to make the equation useful, we have to be able to determine the matrix elements $\langle A^t_s \rangle_n$.

While in the main text we focus our attention on KdV charges, we have not used any of their particular properties, and (3.1) applies to other conserved charges with the appropriate modifications. E.g. flavor symmetry charges $Q$ (namely $s = 0$) remain fixed: the equation gives $\partial_\lambda \langle Q \rangle_n = 0$ because $A^u_0 = 0$. A more general framework for charges is worked out in Appendix G. However, we leave for future work the incorporation of supersymmetry into the algebraic framework used in this paper.

A note of caution is in order: we have yet to fix the ambiguities corresponding to the mixing of conserved charges discussed below (2.16). Without this, (3.1) is just valid for one choice of KdV charges. In Lorentz invariant theories ($u = -t$) we can use spin to prevent the KdV charges from mixing with each other. It can be checked that (2.16) and hence (3.1) respects spin. In fact, when the seed theory is Lorentz-invariant, we can use Lorentz-invariance even for $u \neq -t$, by assigning a spin to the coupling $\lambda$. This spurion analysis is performed in Appendix C.

We have analyzed and solved a problem similar to (3.1) for a family of deformations made out of an abelian current and the stress tensor in [57]. There we have also demonstrated that our current tools are inadequate to determine $\langle A^t_s \rangle_n$ in general. In the rest of this section we focus on the case $X^{1,-1} = T\bar{T}$. In Section 4 we analyze a special case of (3.1) where we can make progress, while we discuss perturbative aspects in Section 4 and in Appendix F.

### 3.2 Evolution under the $T\bar{T}$ deformation

Let us determine the evolution of KdV charges under the $T\bar{T}$ deformation of a Lorentz invariant theory by plugging in $t = 1$, $u = -1$ into (3.1). Using (2.6), we obtain

$$
\partial_\lambda \langle P_s \rangle_n = -\pi s \left( \langle T_{s+1} \rangle_n \langle P_{-1} \rangle_n - \langle \Theta_{s-1} \rangle_n \langle P_1 \rangle_n \right).
\tag{3.2}
$$

We still have to determine the expectation values $\langle T_{s+1} \rangle_n$, $\langle \Theta_{s-1} \rangle_n$. The sum of them gives

$$
\langle T_{s+1} \rangle_n + \langle \Theta_{s-1} \rangle_n = \frac{2\pi}{L} \langle P_s \rangle_n,
\tag{3.3}
$$

---

[3]For momentum ($P = -P_1 + P_{-1}$) the second line of (3.1) gives $\partial_\lambda \langle P \rangle = 0$, as required by momentum quantization.

but the difference, $\langle T_{s+1}\rangle_n - \langle\Theta_{s-1}\rangle_n$ requires additional input. In general the expectation value of the spatial component of a current (which the quantity in question is), $\langle J_x\rangle_n$ does not have a universal expression. In [57] we faced this problem for the case of an internal symmetry current and of $\langle T_{tx}\rangle_n$, and we treated it by introducing background fields. For the case of KdV charges, we found another way to proceed.

Let us first set $s = 1$. From the interpretation of $\langle T_{xx}\rangle_n$ as pressure, we have

$$\langle T_{xx}\rangle_n = -\partial_L E_n. \tag{3.4}$$

Then transforming to complex coordinates,[4]

$$T_{xx} = -\frac{1}{2\pi}\left((T_2 - \Theta_0) - (T_0 - \Theta_{-2})\right), \tag{3.5}$$

and using (3.3) together with the $T_0 = \Theta_0$ valid in Lorentz invariant theories, we read off

$$\langle T_2\rangle_n - \langle\Theta_0\rangle_n = -\pi\left(\partial_L\langle P_{-1} + P_1\rangle_n + \frac{\langle P_{-1} - P_1\rangle_n}{L}\right) = -2\pi\partial_L\langle P_1\rangle_n, \tag{3.6}$$

where in the second equality we used (2.3) and that $\partial_L P_n = -P_n/L$ which follows from momentum quantization, $P_n \in \frac{2\pi\mathbb{Z}}{L}$. Similarly, we get $\langle T_0\rangle_n - \langle\Theta_{-2}\rangle_n = 2\pi\partial_L\langle P_{-1}\rangle_n$.

This motivates us to compute $\partial_L P_s$. In Appendix E we show

$$L\partial_L P_s = \frac{-1}{2\pi}\int dx\left(A_s^1 - A_s^{-1}\right) \quad \text{modulo } [P, \bullet], \tag{3.7}$$

which reduces to the equations above for $s = \pm 1$. Taking diagonal matrix elements and using that $\partial_L\langle P_s\rangle_n = \langle\partial_L P_s\rangle_n$ (valid in eigenstates of $P_s$), we find

$$\partial_L\langle P_s\rangle_n = \frac{-1}{2\pi}\langle A_s^1 - A_s^{-1}\rangle_n = \frac{-s}{2\pi}\left(\langle T_{s+1}\rangle_n - \langle\Theta_{s-1}\rangle_n\right), \tag{3.8}$$

where in the second equality we used (2.6) valid in Lorentz invariant theories.

From (3.8) and (3.3) we can express $\langle T_{s+1}\rangle_n$ and $\langle\Theta_{s-1}\rangle_n$ separately, and plug back their expression into (3.2) to find the flow equation:

$$\boxed{\partial_\lambda\langle P_s\rangle_n = -\pi^2\left(E_n\partial_L\langle P_s\rangle_n + P_n\frac{s\langle P_s\rangle_n}{L}\right).} \tag{3.9}$$

This is our main result. Setting $s = \pm 1$ and using (2.3) we recover the Burgers equation for $E_n$, and the fact that $P_n$ remains undeformed:

$$\partial_\lambda E_n = -\pi^2\left(E_n\partial_L E_n + \frac{P_n^2}{L}\right), \qquad \partial_\lambda P_n = 0, \tag{3.10}$$

where in the second equation we used $\partial_L P_n = -P_n/L$. The KdV charges obey linear equations, (3.9), which take as an input the energy eigenvalue that solves the nonlinear Burgers equation. We provided a hydrodynamical interpretation of these results in the Introduction. We find a similar set of evolution equations in Section 4, where we also show that (3.9) holds even in the absence of Lorentz invariance in the zero-momentum sector ($P_n = 0$) of the theory.

Let us start by solving the equations in two special case. We drop the expectation value symbols and the $n$ subscript to lighten the notation. If we set $P = 0$, the equation simply propagates the initial data $P_s(L)$ along characteristics determined by $E$:

$$E(\lambda, L) = E^{(0)}\left(L - \pi^2\lambda E(\lambda, L)\right),$$
$$P_s(\lambda, L) = P_s^{(0)}\left(L - \pi^2\lambda E(\lambda, L)\right). \tag{3.11}$$

---

[4]We use the same conventions as in [57]. In (A.9) of that paper, we gave this result.

In the conformal case we can solve the equations for any eigenstate. The initial conditions are

$$P_s^{(0)}(L) = \frac{p_s}{L^{|s|}},\tag{3.12}$$

where $p_s$ are numbers only dependent on the state, but not on $L$. The solution of the Burgers equation with this initial data is familiar from the literature:[5]

$$E(\lambda, L) = \frac{1 - \sqrt{1 - 2e\tilde{\lambda} + p^2\tilde{\lambda}^2}}{\tilde{\lambda}L}, \qquad P(\lambda, L) = \frac{p}{L},$$
$$\tilde{\lambda} \equiv \frac{2\pi^2\lambda}{L^2}.\tag{3.13}$$

Once we know the Burgers flow, we can solve for the KdV charges that probe it. In fact we do not have to know the explicit form of the solution, (3.13), to verify that

$$P_s(\lambda, L) = \begin{cases} \frac{p_s}{(p_1)^s} P_1(\lambda, L)^s, & (s > 0), \\ \frac{p_s}{(p_{-1})^{|s|}} P_{-1}(\lambda, L)^{|s|}, & (s < 0), \end{cases}\tag{3.14}$$

solves (3.9), if we use that $P_{\pm 1}(\lambda, L)$ satisfies (3.10).

## 3.3 A check from integrability and concluding comments

Integrable field theories provide a useful testing ground of our results. The $T\bar{T}$ deformation changes the two particle S-matrix of an integrable field theory by a simple CDD factor:

$$S_{T\bar{T}} = \exp\left(-\pi^2\lambda\, m^2 \sinh(\theta_1 - \theta_2)\right) S_0,\tag{3.15}$$

where $m$ is the mass, and $\theta_i$ the rapidities. Plugging this result into the nonlinear integral equation that determines the spectrum gives the deformed spectrum in terms of the initial one. This computation was done for the energy in [5] and extended to KdV charges and other deformations in [58]. Instead of repeating their derivation, we simply copy their equations (4.47) and (4.50) in our notation in (F.1), and here we specialize to the $T\bar{T}$ case (corresponding to taking $u = 1$ in (F.1)). The equation reads

$$\partial_\lambda P_k = \pi^2\left(L'\partial_L P_k - k\,\theta_0' P_k\right)$$
$$L' \equiv P_1 + P_{-1} = -E$$
$$\theta_0' \equiv -\frac{P_1 - P_{-1}}{L} = \frac{P}{L}.\tag{3.16}$$

We recognize that the flow equation is identical to (3.9). This match is a strong check of our results. In Appendix F we discuss in detail their deformations with $u \neq 1$.

Let us comment on the regimes of validity of the different derivations of (3.9). The derivation by [5,58] applies to the sine-Gordon model and minimal model CFTs. The derivation is expected to generalize straightforwardly to any massive integrable model. Our derivation applies to the $T\bar{T}$ deformation of any Lorentz-invariant theory that contains at least one higher spin conserved charge, and hence is more general than that of [5,58]. Our result rules out the possibility that the evolution equation (3.9) is a miracle of some special (integrable) models.[6]

---

[5]The initial data $p_{\pm 1}$ are related to $e$, $p$ by (2.3), i.e. $p_{\pm 1} = -\frac{e \pm p}{2}$. In a CFT $e = 2\pi\left(h + \bar{h} - \frac{c}{12}\right)$ and $p = 2\pi\left(h - \bar{h}\right)$.

[6]Here we use the word integrable in a restrictive sense; by it we mean that the infinitely many conserved charges completely determine the dynamics of the theory. While any two-dimensional CFT has infinitely many KdV conserved charges, this does not make them integrable. CFTs with a semiclassical holographic dual are prime examples of non-integrable theories with infinitely many conserved charges. It is expected that a generic CFT is non-integrable, despite the scarcity of explicit constructions of such theories, see however [77].

A similar relationship holds between the two derivations of the Burgers equation for the $T\bar{T}$-deformed spectrum: the one by [5] applies to the sine-Gordon model and minimal model CFTs, while the one by [4] applies to any Lorentz-invariant field theory.

# 4 Non-Lorentz-invariant deformations

We return to the analysis of (3.1): we study deformations such as $X^{u,-1} = T_{u+1}\bar{T} - \Theta_{u-1}\bar{\Theta}$ (sometimes called $T_{u+1}\bar{T}$ for short) that break Lorentz invariance. Specifically, we write an evolution equation for the spectrum of zero-momentum states under the $X^{1,u} - X^{-1,u}$ deformation. This incidentally implies that our main result (3.9) holds for zero-momentum states even without assuming Lorentz invariance. We then explain why our current methods do not allow writing an evolution equation for general states under these deformations. Finally, we solve the evolution of zero-momentum states and find the asymptotic density of states shows super-Hagedorn growth.

Without Lorentz invariance (for $u \neq \pm 1$) there is no preferred basis in the space of commuting conserved charges, as discussed around (3.1) and in Appendix C. Our results in this section apply to the choice of basis, specified by (2.16), for which (3.1) holds. Importantly, this choice is preferred if the seed theory is a CFT, as we show in Appendix C. This makes it nontrivial to compare our results with those of [58]. What we find in Appendix F is that the two papers describe different deformations, even after accounting for the possible change of basis. It would be interesting to parametrize the ambiguities in our results more completely.

## 4.1 Zero-momentum states

As observed by Cardy [53], Lorentz invariance is not needed to derive the inviscid Burgers equation for energy levels of zero-momentum states under the $T\bar{T}$ deformation. We generalize this to the evolution of all KdV charges of zero-momentum states under the $X^{1,u} - X^{-1,u}$ deformation. This deformation reduces to the usual $T\bar{T}$ deformation both for $u = 1$ and for $u = -1$, and in a CFT it reduces to $T_{u+1}\bar{T}$ for $u > 0$ and $T\bar{T}_{-u+1}$ for $u < 0$.

For the deformation by $X^{1,u} - X^{-1,u}$, (3.1) gives

$$\partial_\lambda \langle P_s \rangle_n = \pi\Big(\langle P_1 - P_{-1} \rangle_n \langle A_s^u \rangle_n - \langle P_u \rangle_n \langle A_s^1 - A_s^{-1} \rangle_n\Big). \tag{4.1}$$

The relation (3.8) $\langle A_s^1 - A_s^{-1} \rangle_n = -2\pi \partial_L \langle P_s \rangle_n$ holds without assuming Lorentz invariance. As always, there is no general way to determine $\langle A_s^u \rangle_n$. For states $|n\rangle$ with zero momentum this issue does not show up since $\langle P_1 - P_{-1} \rangle_n = -P_n = 0$, and one has

$$\partial_\lambda \langle P_s \rangle_n = 2\pi^2 \langle P_u \rangle_n \partial_L \langle P_s \rangle_n \qquad \text{if } P_n = 0. \tag{4.2}$$

For these states, the charge $P_u$ evolves according to the inviscid Burgers equation while all other charges describe probe particles riding the Burgers flow. Taking $u = \pm 1$ we find that our main result (3.9) on the $T\bar{T}$ deformation holds for zero-momentum states even without assuming Lorentz invariance.

We note that (4.2) also describes the deformation $J\bar{T} - J\Theta$, which is a special case of the family of theories analyzed in [57], in the equation we have to make the replacement $\langle P_u \rangle_n \to Q_n/(2\pi)$, with $Q_n$ not evolving with $\lambda$ due to its quantized nature.[7]

It is also interesting to compare with the integrability result (3.16). As we explain in Appendix F the deformation described by integrability techniques is not a deformation by a

---

[7]The $\lambda$-independence of $Q$ is consistent with (4.2): if we set $s = u$, replace $\langle P_u \rangle_n \to \langle Q \rangle_n/(2\pi)$, and use that $\partial_L \langle Q \rangle_n = 0$, we get a consistent equation.

local operator, hence is not in the class we consider. Nevertheless, equations (3.16) and (4.2) surprisingly agree for states that have zero momentum and $\langle P_u \rangle_n = \langle P_{-u} \rangle_n$ (for instance states that are parity-invariant in the seed theory).

Finally, an easy calculation shows that the $X^{1,u} - X^{-1,u}$ and $X^{1,v} - X^{-1,v}$ deformations commute (in the zero-momentum sector), since the following result is symmetric in $u \leftrightarrow v$:

$$
\begin{aligned}
\partial_{\lambda_u} \partial_{\lambda_v} \langle P_s \rangle_n &= 2\pi^2 \big( \partial_{\lambda_u} \langle P_v \rangle_n \partial_L \langle P_s \rangle_n + \langle P_v \rangle_n \partial_L \partial_{\lambda_u} \langle P_s \rangle_n \big) \\
&= 4\pi^4 \big( \langle P_u \rangle_n \partial_L \langle P_v \rangle_n \partial_L \langle P_s \rangle_n + \langle P_v \rangle_n \partial_L \langle P_u \rangle_n \partial_L \langle P_s \rangle_n + \langle P_v \rangle_n \langle P_u \rangle_n \partial_L^2 \langle P_s \rangle_n \big).
\end{aligned}
\tag{4.3}
$$

In [57] we also studied whether deformations commute and we found some cases where they do not. It would be interesting to give a full description of the commutators of different $X^{tu}$ deformations.

## 4.2 General states

The evolution equation (3.1) for KdV charges under an $X^{tu}$ deformation involves expectation values of operators $A_s^u$. Crucially, these $\langle A_s^u \rangle_n$ cannot be determined from the KdV charges $\langle P_k \rangle_n$.

In a CFT, one checks for instance that

$$
A_3^3 = 4 \mathord{:} T^3 \mathord{:} - \frac{c+2}{2} \mathord{:} (\partial T)^2 \mathord{:} + (\text{derivatives})
\tag{4.4}
$$

cannot be written in terms of KdV charges. It is not a linear combination of

$$
T_6 = \mathord{:} T^3 \mathord{:} + \frac{c+2}{12} \mathord{:} (\partial T)^2 \mathord{:}
\tag{4.5}
$$

and of other KdV currents. More stringently, its expectation value in low-level descendants of primary states is not expressible in terms of the eigenvalues of KdV charges $P_1$, $P_3$, $P_5$ (dimensional analysis restricts the set of charges to consider).

The only cases where our main evolution equation (3.1) can be solved with the tools at hand are when the dependence on $\langle A_s^u \rangle_n$ completely drops out. In Section 3 this happened thanks to $A_s^{\pm 1} = \pm s A_{\pm 1}^s$. In (4.1) this happened by restricting to the zero-momentum subsector. It is conceivable that for some seed theories there would be relations between $A_s^u$ with $s, u \neq 0, \pm 1$ and some computable quantities. For instance in a massive free scalar one actually has $A_s^t \simeq T_{s+t} \simeq \Theta_{s+t}$ up to total derivatives. However, one should check whether the relation holds after the deformation.

## 4.3 Evolution of zero-momentum states

The evolution equation (4.2) transports KdV charges along characteristics determined by $P_u$ (to avoid clutter we leave implicit the dependence on $|n\rangle$), so we can simply adapt results (3.11) from the $T\bar{T}$ case and get

$$
\begin{aligned}
P_u(\lambda, L) &= P_u^{(0)}(L + 2\pi^2 \lambda P_u(\lambda, L)), \\
P_s(\lambda, L) &= P_s^{(0)}(L + 2\pi^2 \lambda P_u(\lambda, L)).
\end{aligned}
\tag{4.6}
$$

As for $T\bar{T}$ the solution with CFT initial conditions is much more explicit. We use the same logic as around (3.12). First we set $s = u$, and using the CFT initial conditions (3.12) we find the solution:

$$
P_u(\lambda, L) \equiv \frac{p_u f_u(p_u \tilde{\lambda})}{L^{|u|}}, \qquad \tilde{\lambda} \equiv \frac{2\pi^2 \lambda}{L^{|u|+1}},
\tag{4.7}
$$

where $f_u$ is the unique solution to the polynomial equation

$$(xf_u(x) + 1)^{|u|} f_u(x) = 1 \tag{4.8}$$

that obeys $f_u(0) = 1$.[8] The other KdV charges probe this flow, and they are given by

$$P_s(\lambda, L) = \frac{p_s}{L^{|s|}} f_u(p_u\tilde{\lambda})^{|s/u|} . \tag{4.9}$$

The $J\bar{T} - J\Theta$ deformation ($u = 0$) has to be treated separately, and the solution of (4.2) is

$$P_s(\lambda, L) = \frac{p_s}{(L + \pi\lambda Q_n)^{|s|}} \quad \text{for } u = 0 . \tag{4.10}$$

Note that we get a divergence for $\lambda = -L/(\pi Q_n)$, which is the analog of the branch point that we found for $u \neq 0$, see footnote 8. For the special case of $s = \pm 1$ this result agrees with what was found for the energy spectrum in [57] with very different methods (see also [34]).[9] We take this agreement as a check of both the computations presented in this section and the methods of [57].

## 4.4 The density of states

It is particularly interesting to consider the asymptotic behavior of the spectrum. For that we need to solve (4.8) for $x \to +\infty$,[10] where we get $f_u(x) = x^{-|u|/(|u|+1)} + \ldots$, which for $p_u\tilde{\lambda} \gg 1$ gives

$$P_s(\lambda, L) = \frac{p_s}{L^{|s|}(p_u\tilde{\lambda})^{|s|/(|u|+1)}} + \ldots . \tag{4.11}$$

For $p_u\tilde{\lambda}$ negative enough (see footnote 8) we formally get a complex solution, a familiar behavior from the study of $T\bar{T}$.

In the CFT, high energy primary states in the zero momentum sector have

$$p_u = (-1)^{(|u|+1)/2} \left(\frac{e}{2}\right)^{|u|} + \ldots , \tag{4.12}$$

where the inconvenient alternating sign ultimately follows from the sign in the decomposition $T(x) = -\frac{4\pi^2}{L^2} \sum_k e^{ikx} \left(L_k - \frac{c}{24}\delta_{k,0}\right)$. To have a real asymptotic spectrum, it follows from the condition $p_u\tilde{\lambda} > 0$ that $\lambda < 0$ for $u = \pm 1, \pm 5, \ldots$ and $\lambda > 0$ for $u = \pm 3, \pm 7, \ldots$.

---

[8]One can write a series solution and recast it as a hypergeometric function

$$f_u(x) = \sum_{j=0}^{\infty} \frac{(-x)^j}{j+1} \binom{(j+1)|u| + j - 1}{j}$$

$$= \frac{|u|}{x(|u|+1)} \left( {}_{|u|}F_{|u|-1} \left( \begin{array}{c} \frac{1}{|u|+1}, \ldots, \frac{|u|-1}{|u|+1}, \frac{-1}{|u|+1} \\ \frac{1}{|u|}, \frac{2}{|u|}, \ldots, \frac{|u|-1}{|u|} \end{array} \middle| -x\frac{(|u|+1)^{|u|+1}}{|u|^{|u|}} \right) - 1 \right),$$

which takes real values for $x > x_{\min} \equiv -|u|^{|u|}/(|u|+1)^{|u|+1}$ and has a branch point at $x = x_{\min}$. Another way to find this branch point is to compute the discriminant of (4.8), when seen as a polynomial of $f_u(x)$. The discriminant is $(-1)^{|u|(|u|-1)/2} x^{|u|^2-1} |u|^{|u|} (x - x_{\min})$, which vanishes at $x_{\min}$, indicating that two solutions collide for this value of $x$. This is the analogue of the square-root singularity in the usual Burgers equation.

[9]To recover this result from the formulas (6.4) of [57], we take $A = 0$ corresponding to the $J\bar{T} - J\Theta$ deformation, then $E = (1/L)(s - C/B) = e/(L + \pi\lambda Q_n)$, where we simply set $g_{J\bar{T}} = -g_{J\Theta} = 1$, and $\ell = \lambda$ and specialize to zero momentum. The very attentive reader will notice that we absorbed an $i$ in the definition of the deformation compared to [57] to make formulas real. In [57] the special $A = 0$ case was not analyzed separately, this was first done in [34].

[10]The equation does not have a real solution for $x \to -\infty$.

Plugging (4.12) into (4.11) for $s = \pm 1$ we get (again for $p_u \tilde{\lambda} \gg 1$)

$$E(\lambda, L) = \frac{2}{L} \left( \frac{e}{2|\tilde{\lambda}|} \right)^{1/(|u|+1)} + \dots . \tag{4.13}$$

In a CFT, we know that the density of primaries is asymptotically [78]

$$\rho_{\text{primary}}(E) \approx \exp \left( \sqrt{\frac{4\pi^2(c-1)}{3} e} \right), \tag{4.14}$$

where we used that $E = \frac{e}{L}$ in the CFT. Expressing $e$ with the energies of the deformed theory from (4.13), we obtain

$$\rho_{\text{primary}}(E) \approx \exp \left( \sqrt{\frac{8\pi^4(c-1)\,|\lambda|}{3 \times 2^{|u|}} E^{(|u|+1)/2}} \right), \tag{4.15}$$

for the appropriate sign of $\lambda$ that depends on the value of $u$ as discussed above. Note that the density of states is now independent of $L$, in stark contrast to the extensive entropy expected in local field theories. For $u = \pm 1$ the above result is the Hagedorn growth of the density of states of the $T\bar{T}$-deformed theory [7, 24]. We expect that the total density of states including spinning primaries and descendants would exhibit the same behavior, with only numerical factors modified.

A generalization is to deform a CFT by a linear combination $\sum_u \lambda_u (X^{1,u} - X^{-1,u})$. Similar calculations[11] lead to

$$\rho_{\text{primary}}(E) \approx \exp \left( \sqrt{\frac{4\pi^2(c-1)e(E)}{3}} \right), \qquad e(E) = 4\pi^2 \sum_u (-1)^{\frac{|u|+1}{2}} \lambda_u \left( \frac{E}{2} \right)^{|u|+1}. \tag{4.16}$$

Different choices of $\lambda_u$ appear to accomodate arbitrarily strong (e.g., doubly exponential) super-Hagedorn growth of the density of states.[12] However, since our results only concern zero-momentum states, they are not sufficient to determine when the deformation remains well-defined: there could be divergences in the sum over $u$ for some states.

In the case of the $J\bar{T} - J\Theta$ deformation the Cardy growth remains, but the central charge is replaced by a charge dependent expression:

$$\rho(E, Q) \approx \exp \left( \sqrt{\frac{4\pi^2 c \left( 1 + \pi \lambda Q/L \right)}{3} EL} \right) \qquad \text{if } \lambda Q > -\frac{L}{\pi}, \tag{4.17}$$

where we took the full density of states, hence the replacement $(c-1) \to c$. This behavior was understood in [34].

The super-Hagedorn growth of the density of states is a novel behavior exhibited by this system. The three systems known to us with such growth of density of states are flat space quantum gravity in $d$ dimensions, which is expected to have an asymptotic density of states $\rho(E) = \exp \left( \# E^{\frac{d-2}{d-3}} \right)$ from black holes; p-branes, whose density of states was found to grow

---

[11] A convenient shortcut goes as follows. Charges are transported along characteristics, specifically $P_s(\lambda, L) = P_s^{(0)} \left( L + 2\pi^2 \sum_u \lambda_u P_u \right)$ as in (4.6). High-energy primary states of the CFT obey (4.12) $P_s^{(0)} \approx (-1)^{(|s|+1)/2} (E^{(0)}/2)^{|s|}$. This relation is transported along characteristics. Now use the definition $e = L' E^{(0)}(L')$ valid for any $L'$ combined with the transport equation to express the initial dimensionless energy $e$ in terms of the deformed energy $E$: this gives $e = \left( L + \sum_u 2\pi^2 (-1)^{(|u|+1)/2} \lambda_u (E/2)^{|u|} \right) E$. Deleting the negligible term $L$ from this expression and plugging into the Cardy growth (4.14) for $e$ gives (4.16).

[12] Even though (4.16) formally allows for depletion of the density of states if $\lambda_u$ is fine tuned, the formula breaks down for those cases due to a Jacobian factor that we neglected, and we expect descendant states to ruin cancellations either way.

as $\rho(E) = \exp\left(\#E^{\frac{2(d-1)}{d}}\right)$ (with $d = p+1$) in the semiclassical approximation in [79–82], and $d$-dimensional $T\bar{T}$-deformed theories, whose density of states was recently found to agree with this by a large-$N$ analysis [19]. We do not suggest that these theories to have much to do with each other. The result (4.15) however provides extra motivation to study the $T_{u+1}\bar{T}_{u+1}$ deformation, as these Lorentz invariant theories may give rise to exotic UV asymptotics, which would manifest itself in a density of states similar to (4.15). A natural guess based on the simple dependence of $\rho(E)$ on $\lambda$ of (4.15) and dimensional analysis for the density of states in these theories is

$$\rho(E) \stackrel{\text{guess}}{\approx} \exp\left(\sqrt{\#c\,|\lambda|}\,E^u\right). \tag{4.18}$$

New ideas will be needed to establish (or rule out) this guess.

## Acknowledgements

We thank Zohar Komargodski, Alex Maloney, Stefano Negro, Roberto Tateo, and the participants of the "$T\bar{T}$ and Other Solvable Deformations of Quantum Field Theories" workshop for discussions. MM is supported by the Simons Center for Geometry and Physics. BLF gratefully acknowledges support from the Simons Center for Geometry and Physics, Stony Brook University at which some of the research for this paper was performed.

## A   The $A_\sigma^s$'s, their collisions and factorization

### A.1   Manipulating $A_\sigma^s$'s

Let us first derive two simple equations. Combining (2.4) and (2.5), we get:

$$[P_\sigma, A_{\pm 1}^s(z,\bar{z})] = [P_{\pm 1}, A_\sigma^s(z,\bar{z})]. \tag{A.1}$$

On the other hand, $[P_\lambda, P_\tau] = 0$ and the Jacobi identity imply that

$$[P_\lambda, [P_\tau, A_\sigma^s(z,\bar{z})]] = [P_\tau, [P_\lambda, A_\sigma^s(z,\bar{z})]]. \tag{A.2}$$

First, we can deduce a symmetry property. To make the derivation easier to parse, above the equal signs we write the relation we use. We repeatedly transpose neighboring subscripts to find

$$[P_{\pm 1}, [P_\tau, A_\sigma^s(z,\bar{z})]] \stackrel{\text{(A.2)}}{=} [P_\tau, [P_{\pm 1}, A_\sigma^s(z,\bar{z})]] \stackrel{\text{(A.1)}}{=} [P_\tau, [P_\sigma, A_{\pm 1}^s(z,\bar{z})]]$$

$$\stackrel{\text{(A.2)}}{=} [P_\sigma, [P_\tau, A_{\pm 1}^s(z,\bar{z})]] \stackrel{\text{(A.1)}}{=} [P_\sigma, [P_{\pm 1}, A_\tau^s(z,\bar{z})]] \stackrel{\text{(A.2)}}{=} [P_{\pm 1}, [P_\sigma, A_\tau^s(z,\bar{z})]]. \tag{A.3}$$

This implies that $[P_\tau, A_\sigma^s(z,\bar{z})]$ and $[P_\sigma, A_\tau^s(z,\bar{z})]$ can at most differ by a constant. Taking the expectation value of both quantities in a joint eigenstate of $(P_\tau, P_\sigma)$ gives zero, thus we conclude from (A.3) that

$$[P_\tau, A_\sigma^s(z)] = [P_\sigma, A_\tau^s(z)]. \tag{A.4}$$

This also shows more generally that $[P_{\sigma_n}, \ldots [P_{\sigma_1}, A_{\sigma_0}^s(z,\bar{z})]\ldots]$ is totally symmetric in the $\sigma_i$.

Second, we can establish that the condition (2.12) is obeyed for $X = X^{tu}$. We work here with the point-splitted version of $X^{tu}$ and return later to the discussion of regulator terms. The key identity is a generalization of (A.4) involving operators $A_{\sigma_j}^{s_j}(z_j, \bar{z}_j)$ at $k$ different points:

$$[P_{[\tau}, A_{\sigma_1}^{s_1} \ldots A_{\sigma_k]}^{s_k}] \stackrel{[A,BC]=[A,B]C+B[A,C]}{=} \sum_{j=1}^{k} A_{[\sigma_1}^{s_1} \cdots A_{\sigma_{j-1}}^{s_{j-1}} [P_\tau, A_{\sigma_j}^{s_j}] A_{\sigma_{j+1}}^{s_{j+1}} \cdots A_{\sigma_k]}^{s_k} \stackrel{\text{(A.4)}}{=} 0, \tag{A.5}$$

in which the $k+1$ subscripts are totally antisymmetrized. Specializing to $k=2$ and $\vec{\sigma}=(1,-1)$ we learn that

$$[P_\tau, A^{s_1}_{[1}A^{s_2}_{-1]}] = [P_1, A^{s_1}_{[\tau}A^{s_2}_{-1]}] + [P_{-1}, A^{s_1}_{[1}A^{s_2}_{\tau]}] \tag{A.6}$$

is a total derivative: this is condition (2.12) for the point-splitted version of $X^{s_1 s_2} = 2(A^{s_1}_{[1}A^{s_2}_{-1]})_{\text{reg}}$. Note that for any product other than $A^{s_1}_{[1}A^{s_2}_{-1]}$ on the left-hand side we would have gotten commutators on the right-hand side beyond just derivatives $[P_{\pm 1}, \dots]$.

## A.2 Collision limits

To go from the point-split equation (A.6) to an equation for $X^{s_1 s_2}$ itself we need to understand collision limits of $A^s_\sigma$ operators. Let us define the point-splitted object (here $z_j$ stands for $(z_j, \bar{z}_j)$)

$$\tilde{\mathcal{X}}^{s_1\dots s_k}_{\sigma_1\dots\sigma_k}(z_1,\dots,z_k) \equiv k! A^{s_1}_{[\sigma_1}(z_1)\cdots A^{s_k}_{\sigma_k]}(z_k). \tag{A.7}$$

We take a derivative with respect to one of the coordinates only (say, the first), keeping implicit the position dependence of each $A^{s_j}_{\sigma_i}(z_j)$ for brevity:

$$[P_{\pm 1}, A^{s_1}_{[\sigma_1}]\cdots A^{s_k}_{\sigma_k]} \stackrel{\text{(A.4)}}{=} [P_{[\sigma_1|}, A^{s_1}_{\pm 1}]A^{s_2}_{|\sigma_2}\cdots A^{s_k}_{\sigma_k]}$$

$$\stackrel{[A,B]C=[A,BC]-B[A,C]}{=} [P_{[\sigma_1|}, A^{s_1}_{\pm 1}A^{s_2}_{|\sigma_2}\cdots A^{s_k}_{\sigma_k]}] - A^{s_1}_{\pm 1}[P_{[\sigma_1}, A^{s_2}_{\sigma_2}\cdots A^{s_k}_{\sigma_k]}] \tag{A.8}$$

$$\stackrel{\text{(A.5) on second term}}{=} [P_{[\sigma_1|}, A^{s_1}_{\pm 1}A^{s_2}_{|\sigma_2}\cdots A^{s_k}_{\sigma_k]}].$$

The notation means that $\sigma_i$ indices (but not $\pm 1$) are antisymmetrized in each term. The result is a sum of $[P_{\sigma_i}, \bullet]$ and we shall call it a $P_\sigma$-commutator. Similarly, derivatives of $A^{s_1}_{[\sigma_1}\cdots A^{s_k}_{\sigma_k]}$ with respect to any of the $z_j$ or $\bar{z}_j$ are $P_\sigma$-commutators. In fact, (A.8) also holds with $\pm 1$ replaced by any $\tau$, but we will not use that observation.

We have just shown that all derivatives of $\tilde{\mathcal{X}}^{s_1\dots s_k}_{\sigma_1\dots\sigma_k}$ are $P_\sigma$-commutators. Let us use the OPE

$$A^{s_1}_{\sigma_1}(z_1)\cdots A^{s_k}_{\sigma_k}(z_k) = \sum_\alpha f_\alpha(z_1-w,\dots)\mathcal{O}^{s_1\dots s_k,\alpha}_{\sigma_1\dots\sigma_k}(w), \tag{A.9}$$

written in a basis of functions $f_\alpha(z_1-w,\dots)$ that includes the constant function $f_0(z_1-w,\dots) = 1$. (Typically one can use monomials $\prod_i(z_i-w)^{\alpha_i}$.) Antisymmetrizing over indices $\sigma_1\dots\sigma_k$, what we have shown above is that

$$\sum_\alpha \nabla f_\alpha(z_1-w,\dots)\mathcal{O}^{s_1\dots s_k,\alpha}_{[\sigma_1\dots\sigma_k]}(w) \tag{A.10}$$

is a $P_\sigma$-commutator, where $\nabla$ denotes the vector of all $z_i$ and $\bar{z}_i$ derivatives. Since $f_\alpha$ form a basis, we learn that each $\mathcal{O}^{s_1\dots s_k,\alpha}_{[\sigma_1\dots\sigma_k]}$ is a $P_\sigma$-commutator except for $\alpha = 0$ (constant coefficient). Hence,

$$\tilde{\mathcal{X}}^{s_1\dots s_k}_{\sigma_1\dots\sigma_k}(z_1,\cdots,z_k) = \mathcal{X}^{s_1\dots s_k}_{\sigma_1\dots\sigma_k}(w) + \sum_{\alpha\neq 0} f_\alpha(z_1-w,\dots)\sum_{i=1}^k [P_{\sigma_i}, O^{s_1\dots s_k,\alpha}_{\sigma_1\dots\sigma_k,i}(w)]. \tag{A.11}$$

The $z$-independent term defines a local operator $\mathcal{X}^{s_1\dots s_k}_{\sigma_1\dots\sigma_k}(w)$, and the other terms are counterterms. Due to freedom in changing basis, which may add constants to the coefficients $f_\alpha$ of the counterterms, the operators $\mathcal{X}$ are only defined up to $P_\sigma$-commutators. Given our construction, the $\mathcal{X}$ are totally antisymmetric in their $\sigma$ indices, but might only be antisymmetric in their

$s$ indices up to $P_\sigma$-commutators.[13] Alternatively we could have written the OPE in a basis of local operators that splits into the subspace spanned by $[P_{\sigma_i}, O(w)]$ for $1 \leq i \leq k$ with local $O(w)$, and a complement of that subspace. This gives another definition of $\mathcal{X}^{s_1...s_k}_{\sigma_1...\sigma_k}$ modulo $P_\sigma$-commutators.

Recall now that we are trying to take the collision limit in (A.5), namely in

$$[P_{[\sigma_0}, \tilde{\mathcal{X}}^{s_1...s_k}_{\sigma_1...\sigma_k]}(z_1, \ldots, z_k)] = 0. \tag{A.12}$$

We get

$$0 \stackrel{(A.11)}{=} (k+1)\big[P_{[\sigma_0}, \mathcal{X}^{s_1...s_k}_{\sigma_1...\sigma_k]}(w)\big] \\ + \sum_{\alpha \neq 0} f_\alpha(z_1 - w, \ldots) \sum_{0 \leq i \neq j \leq k} (-1)^j \big[P_{\sigma_j}, [P_{\sigma_i}, O^{s_1...s_k, \alpha}_{\sigma_0...\sigma_{j-1}\sigma_{j+1}\sigma_k, i'}(w)]\big], \tag{A.13}$$

where $i' = i - 1$ if $i < j$ and $i' = i$ otherwise so that the $i'$-th subscript of O is $\sigma_i$. One could hope to use the symmetry $[P_{\sigma_j}, [P_{\sigma_i}, O]] = [P_{\sigma_i}, [P_{\sigma_j}, O]]$ to get terms with $i \leftrightarrow j$ to cancel, but this would require the corresponding O operators to be the same, which they are a priori not. Instead, we notice that the equation takes the form of a linear relation between $f_0(z_1 - w, \ldots)$ (because the first term is $z_i$-independent) and $f_\alpha$ with $\alpha \neq 0$. Since these form a basis by assumption, all of their coefficients in the linear relation vanish. In particular we have established the following symmetry for the collision limits:

$$[P_{[\sigma_0}, \mathcal{X}^{s_1...s_k}_{\sigma_1...\sigma_k]}(w)] = 0. \tag{A.14}$$

This then establishes that (A.5) holds in the coincident point limit with regulator terms included. As explained around (A.5), this establishes that the $X^{tu}$ deformation preserves the KdV charges through the key condition (2.12).

## A.3  Factorization of matrix elements

In this appendix we show the factorization property of the composite operators $\mathcal{X}$ in diagonal matrix elements between energy eigensates. We work in a basis of states $|n\rangle$ in which all charges $P_s$ are diagonal. We assume that the theory has a *non-degenerate spectrum*, namely that each joint eigenspace of *all* the charges $P_s$ is one-dimensional. This is a much weaker assumption than the assumption in [1] that the *energy* spectrum is non-degenerate. (For instance CFTs have a highly degenerate energy spectrum.)

Consider basis states $|n\rangle$ and $|n'\rangle$ of equal $P_\sigma$ for some $\sigma$. The $\langle n|\bullet|n'\rangle$ matrix element of (A.4), namely $[P_\tau, A^s_\sigma] = [P_\sigma, A^s_\tau]$, gives

$$\big(\langle P_\tau\rangle_n - \langle P_\tau\rangle_{n'}\big)\langle n|A^s_\sigma|n'\rangle = 0, \tag{A.15}$$

where $\langle P_\tau\rangle_n$ denotes $\langle n|P_\tau|n\rangle$. From the nondegeneracy assumption we deduce that $A^s_\sigma$ is diagonal in this sector. The argument applies likewise to the point splitted $\mathcal{X}$ operator $\tilde{\mathcal{X}}^{s_1...s_k}_{\sigma_1...\sigma_k}$ with a slight modification: $[P_\tau, \mathcal{X}]$ is a sum of commutators $[P_{\sigma_i}, \bullet]$ so we need to restrict $\tilde{\mathcal{X}}$ to a subspace of fixed $P_{\sigma_i}$ for all $i$. Altogether,

$$A^s_\sigma \text{ restricted to fixed } P_\sigma \text{ is diagonal,}$$
$$\tilde{\mathcal{X}}^{s_1...s_k}_{\sigma_1...\sigma_k} \text{ restricted to fixed } P_{\sigma_1}, \ldots, P_{\sigma_k} \text{ is diagonal.} \tag{A.16}$$

---

[13]Relatedly, when point-splitting we only showed that the OPE is regular (up to $P_\sigma$-commutators) when antisymmetrizing the $\sigma_i$: antisymmetrizing the $s_i$ instead may not give a well-defined operator.

Next, insert a complete set of states in a diagonal matrix element of $\tilde{\mathcal{X}}^{s_1\ldots s_k}_{\sigma_1\ldots\sigma_k}$:

$$\langle n|\tilde{\mathcal{X}}^{s_1\ldots s_k}_{\sigma_1\ldots\sigma_k}|n\rangle = k! \sum_m \langle n|A^{s_1}_{[\sigma_1}|m\rangle\langle m|A^{s_2}_{\sigma_2}\ldots A^{s_k}_{\sigma_k]}|n\rangle\,. \tag{A.17}$$

Then consider one of the off-diagonal terms ($m \neq n$) and let $P_\tau$ be one of the charges for which $P_\tau(n) \neq P_\tau(m)$. Such a charge exists by our non-degeneracy assumption. Then we can perform a calculation very similar to (A.8) but using additionally that $[P_\sigma, |m\rangle\langle m|] = \langle P_\sigma\rangle_m|m\rangle\langle m| - |m\rangle\langle m|\langle P_\sigma\rangle_m = 0$. We find

$$
\begin{aligned}
&\left(\langle P_\tau\rangle_n - \langle P_\tau\rangle_m\right)\langle n|A^{s_1}_{[\sigma_1}|m\rangle\langle m|A^{s_2}_{\sigma_2}\ldots A^{s_k}_{\sigma_k]}|n\rangle\\
&= \quad \langle n|\left[P_\tau, A^{s_1}_{[\sigma_1}\right]|m\rangle\langle m|A^{s_2}_{\sigma_2}\ldots A^{s_k}_{\sigma_k]}|n\rangle\\
&\overset{(A.4)}{=} \langle n|\left[P_{[\sigma_1|}, A^{s_1}_\tau\right]|m\rangle\langle m|A^{s_2}_{|\sigma_2}\ldots A^{s_k}_{\sigma_k]}|n\rangle\\
&\overset{(A.5)}{=} \langle n|\left[P_{[\sigma_1|}, \left(A^{s_1}_\tau|m\rangle\langle m|A^{s_2}_{|\sigma_2}\ldots A^{s_k}_{\sigma_k]}\right)\right]|n\rangle = 0\,.
\end{aligned}
\tag{A.18}
$$

Therefore the sum (A.17) above restricts to $|m\rangle = |n\rangle$. An induction on $k$ shows a factorization property generalizing those for $X^{tu}$ proven in [1,4]:

$$\langle n|\tilde{\mathcal{X}}^{s_1\ldots s_k}_{\sigma_1\ldots\sigma_k}|n\rangle = k!\,\langle n|A^{s_1}_{[\sigma_1}|n\rangle\langle n|A^{s_2}_{\sigma_2}|n\rangle\cdots\langle n|A^{s_k}_{\sigma_k]}|n\rangle\,. \tag{A.19}$$

Note that we have omitted the positions of these operators because derivatives with respect to these positions vanish in diagonal matrix elements. We can now take the coincident point limit in $\tilde{\mathcal{X}}$: the regulator (and finite but ambiguous) terms of the form $[P_{\sigma_i}, \bullet]$ drop out in diagonal matrix elements. Hence

$$\langle n|\mathcal{X}^{s_1\ldots s_k}_{\sigma_1\ldots\sigma_k}|n\rangle = \langle n|\tilde{\mathcal{X}}^{s_1\ldots s_k}_{\sigma_1\ldots\sigma_k}|n\rangle\,, \tag{A.20}$$

which combined with (A.19) concludes the proof of factorization.

Zamolodchikov in [1] proved the factorization of diagonal matrix elements of $T\bar{T} - \Theta\bar{\Theta}$ only in states that have no energy and momentum degeneracy. An improvement here is that the equation holds for all states whose degeneracy can be lifted by any set of commuting local conserved charges.[14]

# B  Proof of (2.6) in Lorentz-invariant theories

By construction, $A^s_1 = T_{s+1}$ and $A^s_{-1} = -\Theta_{s-1}$. We show here that in Lorentz-invariant theories the operators $A^{\pm 1}_s$ are given by (2.6), namely

$$A^1_s = s\,T_{s+1} \quad \text{and} \quad A^{-1}_s = s\,\Theta_{s-1}\,, \tag{B.1}$$

provided one suitably improves the symmetry current $(T_{s+1}, \Theta_{s-1})$ by adding a total derivative $\left([P_1, U^s], -[P_{-1}, U^s]\right)$ with $U^s$ given later in (B.14). A surprising side-effect of (B.1) is that it fixes a preferred choice of improvements for all higher-spin currents ($s \neq 0, 1, -1$), because $A^{\pm 1}_s$ are not affected by improvements of $(T_{s+1}, \Theta_{s-1})$. Improvements are discussed in detail in Appendix C.

---

[14]For example, consider a theory with flavor symmetry $\mathfrak{su}(2)$ and consider an irreducible representation $R$ of $\mathfrak{su}(2)$ inside the Hilbert space. Our reasoning shows the factorization property for eigenstates of $i\sigma_3 \in \mathfrak{su}(2)$, but also by symmetry for eigenstates of any other element of $\mathfrak{su}(2)$. How can the non-linear property of factorization hold for all these linearly-related states in $R$ at the same time? The key is that $T$ and $\bar{T}$ and $T\bar{T}$ commute with $\mathfrak{su}(2)$ hence are multiples of the identity when acting on $R$.

Let us first derive a consequence of (B.1). These relations can be stated as $sA_{\pm 1}^s = \pm A_s^{\pm 1}$. Combined with (A.4) $[P_u, A_t^s] = [P_t, A_u^s]$ we get

$$[P_{\pm 1}, sA_t^s] = [P_t, sA_{\pm 1}^s] = [P_t, \pm A_s^{\pm 1}] = \text{(same with } s \leftrightarrow t),\tag{B.2}$$

so derivatives of the difference $sA_t^s - tA_s^t$ vanish. This difference is thus a multiple of the identity, hence must be zero unless its spin $s + t$ is the same as that of the identity operator. In that case $(s + t = 0)$ the multiple of the identity can be absorbed into the definition of $A_{\pm s}^{\mp s}$, for instance by normalizing their ground state expectation value to zero. Altogether, we conclude

$$sA_t^s = tA_s^t.\tag{B.3}$$

It remains to prove (B.1). For $s = \pm 1$, (B.1) is immediate. Our strategy for other spins is to show that $A_s^1 - sT_{s+1}$ and $A_s^{-1} - s\Theta_{s-1}$ have vanishing $\partial_x$ derivative, as we state in (B.15) and (B.22). Then these local operators must be multiples of the identity, hence vanish because their spin is non-zero $(s \pm 1 \neq 0)$. In the special case $s = 0$ one determines $A_0^{\pm 1} = 0$ through their derivatives $[P_{\pm 1}, A_0^{\pm 1}] = [P_0, A_{\pm 1}^{\pm 1}] = 0$, where we used that a flavor symmetry charge $P_0$ commutes with all stress tensor components $A_{\pm 1}^{\pm 1}$. Henceforth we focus on spins $s \neq 0, -1, 1$.

Throughout our proof of (B.1) we write equal-time commutators of local operators as[15]

$$[A(x), B(y)] = \sum_{n \geq 0} \mathcal{O}_n(A, B; y)\partial_x^n \delta(x - y).\tag{B.4}$$

Notice for instance that $\mathcal{O}_0(A, B; y)$ and $-\mathcal{O}_0(B, A; y)$ differ by derivatives since

$$\mathcal{O}_0(A, B; y) = \int dx\, [A(x), B(y)] = -\int dx\, [B(y), A(x)] = -\sum_{n \geq 0} \partial_y^n \mathcal{O}_n(B, A; y).\tag{B.5}$$

## B.1 Computing some derivatives

First we work out

$$\begin{aligned}
\partial_x A_s^1(x) &= i[P_1 - P_{-1}, A_s^1(x)]\\
&= i[P_s, T_2(x) + \Theta_0(x)]\\
&= \frac{i}{2\pi} \int dy\, [T_{s+1}(y) + \Theta_{s-1}(y), T_2(x) + \Theta_0(x)]\\
&\stackrel{(B.5)}{=} -\frac{i}{2\pi} \sum_{n \geq 0} \partial_x^n \mathcal{O}_n(T_2 + \Theta_0, T_{s+1} + \Theta_{s-1}, x)
\end{aligned}\tag{B.6}$$

and likewise

$$\partial_x A_s^{-1} = -\frac{i}{2\pi} \sum_{n \geq 0} \partial_x^n \mathcal{O}_n(T_0 + \Theta_{-2}, T_{s+1} + \Theta_{s-1}).\tag{B.7}$$

All terms except the $n = 0$ ones are manifestly $x$-derivatives. Let us check the $n = 0$ terms also are:

$$\begin{aligned}
\frac{-i}{2\pi} \mathcal{O}_0(T_2 + \Theta_0, T_{s+1} + \Theta_{s-1}) &= -i[P_1, T_{s+1} + \Theta_{s-1}] = -i[P_1 - P_{-1}, T_{s+1}] = -\partial_x T_{s+1},\\
\frac{-i}{2\pi} \mathcal{O}_0(T_0 + \Theta_{-2}, T_{s+1} + \Theta_{s-1}) &= -i[P_{-1}, T_{s+1} + \Theta_{s-1}] = i[P_1 - P_{-1}, \Theta_{s-1}] = \partial_x \Theta_{s-1}.
\end{aligned}\tag{B.8}$$

Thus, when restricted to $n \geq 1$, the sums in (B.6) and (B.7) give $\partial_x(A_s^1 + T_{s+1})$ and $\partial_x(A_s^{-1} - \Theta_{s-1})$. From these we want to subtract derivatives of $(s + 1)T_{s+1}$ and $(s - 1)\Theta_{s-1}$ respectively.

---

[15]We sometimes denote $\mathcal{O}_n(A, B)$ without specifying the point $y$ when that point is clear from context.

To make factors of spin appear, we consider commutators with the spin operator $S$ acting by rotations around the point $x$. By definition,[16]

$$S = i \int dy \, (y-x) T_{tt}(y), \tag{B.9}$$

so we can express $[S, A(x)]$ for any local operator $A(x)$ in terms of the commutator $[T_{tt}(y), A(x)]$:

$$[S, A(x)] = i \int dy \, (y-x) \sum_{n \geq 0} \mathcal{O}_n(T_{tt}, A; x) \partial_y^n \delta(y-x) = -i \mathcal{O}_1(T_{tt}, A; x). \tag{B.10}$$

From the fact that $T_{s+1}$ and $\Theta_{s-1}$ have spins $s \pm 1$ we learn that

$$\begin{aligned}(s+1) T_{s+1} &= -i \mathcal{O}_1(T_{tt}, T_{s+1}), \\ (s-1) \Theta_{s-1} &= -i \mathcal{O}_1(T_{tt}, \Theta_{s-1}).\end{aligned} \tag{B.11}$$

We obtain that derivatives of $A_s^1 - s T_{s+1}$ and $A_s^{-1} - s \Theta_{s-1}$ are quite complicated:

$$\begin{aligned}\partial_x(A_s^1 - s T_{s+1}) &= i \partial_x \mathcal{O}_1(T_{tt}, T_{s+1}) - \frac{i}{2\pi} \sum_{n \geq 1} \partial_x^n \mathcal{O}_n(T_2 + \Theta_0, T_{s+1} + \Theta_{s-1}), \\ \partial_x(A_s^{-1} - s \Theta_{s-1}) &= i \partial_x \mathcal{O}_1(T_{tt}, \Theta_{s-1}) - \frac{i}{2\pi} \sum_{n \geq 1} \partial_x^n \mathcal{O}_n(T_0 + \Theta_{-2}, T_{s+1} + \Theta_{s-1}).\end{aligned} \tag{B.12}$$

## B.2 Improvement

It is not immediately obvious how to absorb right-hand sides into an improvement of $(T_{s+1}, \Theta_{s-1})$. Because $2\pi T_{tt} = T_2 + \Theta_0 + T_0 + \Theta_{-2}$, the sum of these equations simplifies and gives second derivatives and higher:

$$\partial_x(A_s^1 + A_s^{-1} - s T_{s+1} - s \Theta_{s-1}) = -i \sum_{n \geq 2} \partial_x^n \mathcal{O}_n(T_{tt}, T_{s+1} + \Theta_{s-1}). \tag{B.13}$$

This is precisely as expected because the time component $T_{s+1} + \Theta_{s-1}$ of a current is shifted by a space derivative upon improvements. For $s \neq 0$ the right-hand side of (B.13) is absorbed by using the following improved current (in the main text we drop the hats)[17]

$$\begin{aligned}\hat{T}_{s+1} &= T_{s+1} + [P_1, U^s], \qquad \hat{\Theta}_{s-1} = \Theta_{s-1} - [P_{-1}, U^s], \\ U^s &:= \frac{1}{s} \sum_{n \geq 2} \partial_x^{n-2} \mathcal{O}_n(T_{tt}, T_{s+1} + \Theta_{s-1}) \quad \text{and} \quad U^0 := 0.\end{aligned} \tag{B.14}$$

Explicitly,

$$\partial_x(A_s^1 + A_s^{-1} - s \hat{T}_{s+1} - s \hat{\Theta}_{s-1}) = 0. \tag{B.15}$$

---

[16]We left the point of origin $x$ implicit in our notation for $S$. $S$ is the charge corresponding to the rotation current $j_\mu(y) \equiv \epsilon^{\alpha\beta}(y-x)_\alpha T_{\beta\mu}$ that is conserved by virtue of the symmetry of the stress tensor. Since the coordinates $x$, $y$ are not well-defined on the cylinder, the expression we gave for $S$ only makes sense locally, but our calculations are local so doing them on the plane would be equivalent.

[17]For $s = \pm 1$ the left-hand side of (B.13) vanishes by construction, so the right-hand side must vanish. This is difficult to prove by direct calculations.

### B.3 Space component

Next we prove the analogous equation for the space components, namely with the signs of $A_s^{-1}$ and $\hat{\Theta}_{s-1}$ flipped. We first compute the improvement term in $sT_{s+1} - s\Theta_{s-1}$, namely $s[P_1 + P_{-1}, U^s]$. It involves the operator $[P_1 + P_{-1}, \mathcal{O}_n(T_{tt}, T_{s+1} + \Theta_{s-1})]$, which, by definition of $\mathcal{O}_n$, is the $n$-th term in the following commutator

$$\left[P_1 + P_{-1}, [T_{tt}(x), T_{s+1}(y) + \Theta_{s-1}(y)]\right] = \sum_{n\geq 0}[P_1 + P_{-1}, \mathcal{O}_n(T_{tt}, T_{s+1} + \Theta_{s-1}; y)]\partial_x^n \delta(x-y). \tag{B.16}$$

Applying the Jacobi identity and the conservation equations for $T_{t\mu}$ and $T_{s+1} \pm \Theta_{s-1}$ gives

$$\left[\partial_x T_{tx}(x), T_{s+1}(y) + \Theta_{s-1}(y)\right] - i\left[T_{tt}(x), \partial_y(T_{s+1}(y) - \Theta_{s-1}(y))\right]. \tag{B.17}$$

The space derivatives $\partial_x$ and $\partial_y$ can be pulled out of the commutators, which can then both be expanded as $\sum_{n\geq 0}\mathcal{O}_n(\ldots;y)\partial_x^n \delta(x-y)$ with appropriate arguments. Moving the derivatives back into the sum gives

$$\sum_{n\geq 0}\left(\left(\mathcal{O}_n(T_{tx}, T_{s+1} + \Theta_{s-1}; y) + i\mathcal{O}_n(T_{tt}, T_{s+1} - \Theta_{s-1}; y)\right)\partial_x^{n+1}\delta(x-y) \atop -i\partial_y\mathcal{O}_n(T_{tt}, T_{s+1} - \Theta_{s-1}; y)\partial_x^n \delta(x-y)\right). \tag{B.18}$$

Equating coefficients of $\partial_x^n \delta(x-y)$ in (B.16) and (B.18) teaches us that for $n \geq 1$

$$[P_1 + P_{-1}, \mathcal{O}_n(T_{tt}, T_{s+1} + \Theta_{s-1})]$$
$$= \mathcal{O}_{n-1}(T_{tx}, T_{s+1} + \Theta_{s-1}) + i\mathcal{O}_{n-1}(T_{tt}, T_{s+1} - \Theta_{s-1}) - i\partial_x \mathcal{O}_n(T_{tt}, T_{s+1} - \Theta_{s-1}). \tag{B.19}$$

We conclude that

$$s[P_1 + P_{-1}, U^s] = \sum_{n\geq 2}\partial_x^{n-2}[P_1 + P_{-1}, \mathcal{O}_n(T_{tt}, T_{s+1} + \Theta_{s-1})]$$
$$= i\mathcal{O}_1(T_{tt}, T_{s+1} - \Theta_{s-1}) + \sum_{n\geq 1}\partial_x^{n-1}\mathcal{O}_n(T_{tx}, T_{s+1} + \Theta_{s-1}). \tag{B.20}$$

Returning to (B.12) and using $T_{tx} = T_{xt}$ we work out

$$\partial_x(A_s^1 - A_s^{-1} - sT_{s+1} + s\Theta_{s-1})$$
$$= i\partial_x\mathcal{O}_1(T_{tt}, T_{s+1} - \Theta_{s-1}) + \sum_{n\geq 1}\partial_x^n \mathcal{O}_n(T_{xt}, T_{s+1} + \Theta_{s-1}) = \partial_x(s[P_1 + P_{-1}, U^s]), \tag{B.21}$$

namely

$$\partial_x(A_s^1 - A_s^{-1} - s\hat{T}_{s+1} + s\hat{\Theta}_{s-1}) = 0. \tag{B.22}$$

Since if the space derivative of an operator with spin vanishes, it must be the zero operator, (B.15) and (B.22) conclude the proof of (2.6).

## C  Ambiguities

In this Appendix we collect results about ambiguities that we encountered in our derivation. First we present four ambiguities, the most problematic being the ambiguity in choosing the basis of conserved charges. For a Lorentz-invariant seed theory we use Lorentz invariance and a spurion analysis to partly resolve this basis ambiguity. For a CFT seed, dimensional analysis mostly eliminates the remaining basis ambiguity. In cases where we are eventually unable to resolve some of the ambiguity, our equations are only valid for the specific choice of basis that we prescribe.

## C.1  Four ambiguities

Conserved currents are only defined up to improvement transformations. Under an improvement $(T_{s+1}, \Theta_{s-1}) \to (T_{s+1} + \partial \mathcal{O}^s, \Theta_{s-1} + \bar{\partial} \mathcal{O}^s)$, we get using (2.4) that $A^s_\sigma \to A^s_\sigma + i[P_\sigma, \mathcal{O}^s]$. Let us now take antisymmetric combinations of the $A^s_\sigma$'s that define the operator $\mathcal{X}^{s_1 \dots s_k}_{\sigma_1 \dots \sigma_k}$ modulo $P_\sigma$-commutators (see Appendix A.2). Under an improvement the point-splitted operator is shifted as

$$\tilde{\mathcal{X}}^{s_1 \dots s_k}_{\sigma_1 \dots \sigma_k} \to \tilde{\mathcal{X}}^{s_1 \dots s_k}_{\sigma_1 \dots \sigma_k} + \sum_{i=0}^{k} k! A^{s_1}_{[\sigma_1} \dots A^{s_{i-1}}_{\sigma_{i-1}} [P_{\sigma_i}, \mathcal{O}^{s_i}] A^{s_{i+1}}_{\sigma_{i+1}} \dots A^{s_k}_{\sigma_k]}, \tag{C.1}$$

where each term in the sum can be rewritten as $[P_{\sigma_i}, \dots]$ using (A.5). The change in the collision $\mathcal{X}^{s_1 \dots s_k}_{\sigma_1 \dots \sigma_k}$ due to improvements can thus be absorbed into the regulator terms ($P_\sigma$-commutators), as claimed below (2.7).

Note that the ambiguity in the choice of these regulator terms drops out from diagonal matrix elements in joint eigenstates of KdV charges, since $\langle n |[P_\sigma, \mathcal{O}]| n \rangle = 0$. In fact, under an improvement none of the expectation values on either side of the factorization property (A.19)–(A.20) are affected:

$$\langle n | \mathcal{X}^{s_1 \dots s_k}_{\sigma_1 \dots \sigma_k} | n \rangle = k! \, \langle n | A^{s_1}_{[\sigma_1} | n \rangle \langle n | A^{s_2}_{\sigma_2} | n \rangle \cdots \langle n | A^{s_k}_{\sigma_k]} | n \rangle . \tag{C.2}$$

There is a trivial ambiguity in the definition of $A^s_\sigma$, the shift by multiples of the identity: $A^s_\sigma \to A^s_\sigma + a^s_\sigma \mathbb{1}$. Because (2.5) fixes $a^s_{\pm 1} = 0$, the ambiguity does not affect $X^{st} = \mathcal{X}^{tu}_{-1,1}$. However, it changes $\mathcal{X}^{s_1 \dots s_k}_{\sigma_1 \dots \sigma_k}$ by mixing it with combinations of $\mathcal{X}$ of fewer indices. The only case relevant to us is $\mathcal{X}^{tu}_{s, \pm 1} \to \mathcal{X}^{tu}_{s, \pm 1} + 2a^{[t}_s A^{u]}_{\pm 1}$: the variation (2.16) of $P_s$ under the $X^{tu}$ deformation is constructed from it and we get

$$\partial_\lambda P_s \to \partial_\lambda P_s - \pi a^t_s P_u + \pi a^u_s P_t . \tag{C.3}$$

This mixing of charges is a special case of the ambiguities discussed next.

Finally, we focus on an ambiguity that is not easily resolved. The algebra of local conserved charges is in general non-abelian (for instance in case of non-abelian flavor symmetry); for our purposes we need to choose a maximal commuting subalgebra that includes the Hamiltonian and momentum. Within this subalgebra, we still have to choose a basis. While any function of the charges $P_s$ is conserved, only their linear combinations plus shifts by the identity must derive from a local conserved current. Let us implement the change $\delta P_s = \sum_t M_{st} P_t + \frac{LN_s}{2\pi} \mathbb{1}$, with $\delta P_{\pm 1} = 0$ (so $M_{\pm 1, t} = N_{\pm 1} = 0$) to respect momentum quantization and the fact that our deformations are always specified by how they act on the energy, with no ambiguity. It shifts local operators as follows:

$$\delta A^s_\sigma = \sum_t M_{st} A^t_\sigma + \sum_t M_{\sigma t} A^s_t + N_s \delta_{\sigma, 1} \mathbb{1} ,$$

$$\delta \mathcal{X}^{s_1 s_2}_{\sigma_1 \sigma_2} = \sum_t \left( M_{s_1 t} \mathcal{X}^{t s_2}_{\sigma_1 \sigma_2} + M_{s_2 t} \mathcal{X}^{s_1 t}_{\sigma_1 \sigma_2} + M_{\sigma_1 t} \mathcal{X}^{s_1 s_2}_{t \sigma_2} + M_{\sigma_2 t} \mathcal{X}^{s_1 s_2}_{\sigma_1 t} \right) \tag{C.4}$$

$$+ 2 \left( N_{s_1} \delta_{1, [\sigma_1} A^{s_2}_{\sigma_2]} - N_{s_2} \delta_{1, [\sigma_1} A^{s_1}_{\sigma_2]} \right) ,$$

where the shift of $A^s_\sigma$ is a particular choice that preserves (2.5). There are other satisfactory choices, as discussed around (C.3).

This basis ambiguity enters as follows in the story presented in the main text. The definition of $Y_{\pm 1}$ in (2.12) is ambiguous by the addition of conserved currents, and this leads to a freedom of adding a linear combination of conserved currents to (2.16). We consider below various conditions on the seed theory or on the deformation and determine how much they reduce the ambiguity. This may be useful when comparing our results to other approaches, as such approaches may only respect some of the conditions that we use to uniquely characterize our choice of deformation.

## C.2 Lorentz invariance, spurions and dimensional analysis

Consider first the Lorentz-preserving $X^{u,-u}$ deformation of a relativistic theory. One may not add multiples of the identity to any charge: indeed, the identity could only be added to current components of spin 0, namely $\Theta = \Theta_0$ and $\bar{\Theta} = T_0$, but these are fixed by $\delta P_{\pm 1} = 0$. In addition, one may only linearly combine currents of the same spin, namely shift $\partial_\lambda P_s$ by $\alpha_s(\lambda) P_s$ for some coefficients $\alpha_s$ (more generally a combination of all charges of the same spin). If the seed theory is a CFT, dimensional analysis eliminates the ambiguity because it only allows a singular $\alpha_s(\lambda) \sim 1/\lambda$. For a massive theory, $\alpha_s$ can depend nontrivially on the dimensionless combination of the mass scale $\mu$ of the seed theory and the irrelevant coupling $\lambda$. In the absence of a nonabelian charge algebra, no physical principle forbids such rescaling, but there is a minimal choice (2.16) that we employ in this paper.[18] In the $T\bar{T}$ case it is also the natural definition of charges that emerges in the integrability context in [5,58]. If we have a nonabelian algebra, as it is the case for a CFT seed theory, we cannot rescale the different generators arbitrarily as that would violate the commutation relations. The choice made in (2.16) is compatible with the preservation of the algebra as shown in Appendix G. In summary, equation (3.9) giving the evolution of KdV charges under the $T\bar{T}$ flow is unambiguous for a CFT seed, and otherwise its only ambiguity is to scale each KdV charges. This ambiguity is frozen by our choice (2.16).

It is still worth contemplating how easy would it be to recognize the evolution considered in this paper, if we were handed the spectrum of the theory with a different choice of rescaling. Since the rescaling acts the same way on each eigenvalue, the ratio of two eigenvalues is unambiguous, and it would readily lead to the identification of the deformation and the rescaling used.

Next, consider a relativistic seed theory, but deform it by an arbitrary $X^{tu}$. The key to using Lorentz-invariance of the original theory is to promote the coupling $\lambda$ to a background field (also called a spurion) that has spin $-t - u$, so that the action is deformed by the Lorentz-invariant combination $\int d^2 x\, \lambda X^{tu}$. To illustrate how the spurion helps, note that our minimal prescription for $\partial_\lambda P_s$ is an integral of operators $\mathcal{X}^{tu}_{s,\pm 1}$ of spin $s + t + u \pm 1$, consistent with the spins of the current components $\partial_\lambda T_{s+1}$ and $\partial_\lambda \Theta_{s-1}$. Using the same idea, the only mixing ambiguities in the $X^{tu}$ deformation of a relativistic seed are

$$\partial_\lambda P_s \to \partial_\lambda P_s + \sum_{k \geq 1} \alpha_{s,k} \lambda^{k-1} P_{s+k(t+u)}, \tag{C.5}$$

for some coefficients $\alpha_{s,k}$ (more generally one should allow in each term any charge of the same spin as $P_{s+k(t+u)}$). Without further input these ambiguities cannot be eliminated. If the seed is a CFT then we use dimensional analysis: $\lambda$ has dimension $-|t| - |u|$ while $P_s$ has dimension $|s|$. Only terms with $|s + kt + ku| = |s| + k|t| + k|u|$ are dimensionally consistent. This condition means $(s, kt, ku)$ have the same sign or are zero.

In particular, the $X^{tu}$ deformations of a CFT with $tu < 0$ have no ambiguity.

For $tu > 0$ deformations of a CFT (say, $t, u > 0$), $X^{tu}$ vanishes because it is an antisymmetric combination of holomorphic currents. The deformation thus ought to be trivial, but our general prescription (2.16) turns out to mandate a change of basis among holomorphic currents. Indeed, it sets $\partial_\lambda P_s$ to an integral of operators $\mathcal{X}^{tu}_{s,\pm 1}$. For $s < 0$ this vanishes because $A^t_s$ and $A^u_s$ vanish, as $P_s$ is built from a different Virasoro algebra than $P_t$ and $P_u$. For $s > 0$ however, the

---

[18]For deformations other than $T\bar{T}$ this statement must be qualified: (2.16) does not fully define a choice of charges. The ambiguity $A^t_s \to A^t_s + a^t_s \mathbb{1}$ resurfaces. Lorentz-invariance only allows $a^{-s}_s \neq 0$, and because of (B.3) it requires $a^{-s}_s = -a^s_{-s}$. Plugging into (C.3) for the $X^{u,-u}$ deformation we find $\partial_\lambda P_u \to \partial_\lambda P_u + \pi a^{-u}_u P_u$ and $\partial_\lambda P_{-u} \to \partial_\lambda P_{-u} - \pi a^u_{-u} P_{-u}$. This means that (2.16) does not fully define a choice of charges $P_u$ and $P_{-u}$: specifically one could rescale both of them (by the same factor because $a^{-u}_u = -a^u_{-u}$). This caveat does not affect our results: for the $T\bar{T}$ deformation, $a^{-1}_1 = 0$ because of (2.5), so (2.16) fully defines all $\partial_\lambda P_s$.

operator $\mathcal{X}_{s1}^{tu}$ may be non-zero: it is simply a holomorphic conserved current. We see that our general prescription is in this case not a "minimal" choice of how charges are deformed, as one could have taken simply $\partial_\lambda P_s = 0$. (This minimal choice cannot be generalized to non-CFTs.) The spurion and dimensional analysis above simply teaches us that for $s < 0$, $\partial_\lambda P_s = 0$ is not ambiguous, while for $s > 0$ the variation $\partial_\lambda P_s$ has the full ambiguity (C.5). That ambiguity is enough to relate the choice made in (2.16) to the minimal choice.

### C.3   Ambiguities for Section 4

Our spurion analysis (for relativistic seeds) and dimensional analysis (for CFT seeds) extends to linear combinations of deformations by assigning separate spins and dimensions to all of the coupling constants. In particular let us discuss the $X^{1,u} - X^{-1,u}$ deformation of Section 4, taking for definiteness $u > 1$ (the case $u = 1$ is $T\bar{T}$). For the case of a CFT seed we will eliminate the whole ambiguity.

Assume first that we start from a Lorentz-invariant theory. The couplings $\lambda_\pm$ of $X^{\pm 1,u}$ have different spins $-u \mp 1$. A charge $P_s$ can thus be mixed with $\lambda_+^k \lambda_-^l P_{s+k(u+1)+l(u-1)}$ for $k, l \geq 0$. We can now reduce to a single coupling $\lambda_\pm = \pm\lambda$ and write the ambiguity as

$$\partial_\lambda P_s \to \partial_\lambda P_s + \sum_{m \geq 1} \lambda^{m-1} \sum_{k=0}^{m} \alpha_{s,m,k} P_{s+m(u-1)+2k}. \tag{C.6}$$

This ambiguity cannot be eliminated without further assumptions.

For a CFT seed we can eliminate these ambiguities completely. Among ambiguities (C.6) allowed by the spurion analysis, dimensional analysis (where $\lambda$ has dimension $-u - 1$) only allows those with $|s| + m(u+1) = |s + m(u-1) + 2k|$. Using the triangle inequality one has

$$|s + m(u-1) + 2k| \leq |s| + m(u-1) + 2k \leq |s| + m(u+1), \tag{C.7}$$

with equality if and only if $k = m$ and $s \geq 0$. Thus, (C.6) becomes

$$\partial_\lambda P_s \to \partial_\lambda P_s + \sum_{m \geq 1} \lambda^{m-1} \alpha_{s,m} P_{s+m(u+1)} \quad \text{for } s \geq 0. \tag{C.8}$$

Focus on states $|n\rangle$ that start out as primary states in the CFT. Our evolution equation (4.2) preserves $\langle P_s - P_{-s} \rangle_n = 0$. In contrast, any shift (C.8) spoils this because the charges $P_{s+m(u+1)}$ all have positive spins and their expectation values all have different scalings in terms of the state's energy. The condition of preserving $\langle P_s - P_{-s} \rangle_n = 0$ thus characterizes our deformation when the seed is a CFT.

## D   Existence of local currents generating the KdV charges

In this Appendix we show that if $X$ satisfies (2.12) then $P_s$ remains conserved and the integral of a local current. The conservation equation (2.2) in the canonical formalism takes the form

$$0 = [P_{-1}, T_{s+1}] + [P_1, \Theta_{s-1}], \tag{D.1}$$

which we linearize in the coupling of $X$ to obtain

$$0 = [\delta P_{-1}, T_{s+1}] + [P_{-1}, \delta T_{s+1}] + [\delta P_1, \Theta_{s-1}] + [P_1, \delta \Theta_{s-1}]. \tag{D.2}$$

Quantization of the momentum implies $\delta P = 0$, and using $\delta H = \int dy\, X(y)$ together with (2.3) implies that $\delta P_{\pm 1} = -\frac{1}{2} \int dy\, X(y)$, reducing (D.2) to

$$0 = -\frac{1}{2} \int dy\, [X(y), T_{s+1}(x) + \Theta_{s-1}(x)] + [P_{-1}, \delta T_{s+1}(x)] + [P_1, \delta \Theta_{s-1}(x)]. \tag{D.3}$$

The commutator of two local operators can in general be written as

$$[T_{s+1}(x) + \Theta_{s-1}(x), X(y)] = \sum_{n \geq 0} \mathcal{O}_n(y) \partial_x^n \delta(x-y). \tag{D.4}$$

Integrating this commutator over $x$ gives $\mathcal{O}_0(y) = 2\pi[P_s, X(y)]$. In (D.3) we need the integral of this expression in $y$:

$$\int dy \, [X(y), T_{s+1}(x) + \Theta_{s-1}(x)] = -2\pi[P_s, X(x)] - \partial_x \left( \sum_{n \geq 1} \partial_x^{n-1} \mathcal{O}_n(x) \right)$$

$$= -2\pi[P_s, X(x)] + i[P_{-1} - P_1, \sum_{n \geq 1} \partial_x^{n-1} \mathcal{O}_n(x)], \tag{D.5}$$

where we used (2.3). Plugging this result back into (D.3) we see that we can satisfy that equation only if the condition (2.12) is obeyed. Putting (D.5), (2.12), and (D.3) together we get that[19]

$$\delta T_{s+1}(x) = -\pi Y_{-1}(x) + \frac{i}{2} \sum_{n \geq 1} \partial_x^{n-1} \mathcal{O}_n(x),$$

$$\delta \Theta_{s-1}(x) = -\pi Y_1(x) - \frac{i}{2} \sum_{n \geq 1} \partial_x^{n-1} \mathcal{O}_n(x). \tag{D.6}$$

# E   Rescaling space

We show here how KdV charges respond to a rescaling of space. Specifically we show

$$L \partial_L P_s = \frac{-1}{2\pi} \int dx \left( A_s^1 - A_s^{-1} \right) + [P, \mathcal{W}], \tag{E.1}$$

for some nonlocal operator $\mathcal{W}$, which however does not influence diagonal matrix elements in eigenstates, since $\langle n|[P, \bullet]|n \rangle = 0$. One way to reach this equation is to start from $L \partial_L H = -\int dx \, T_{xx}$ and apply the general machinery (2.12) with $X = -T_{xx}$ to determine how KdV charges can be adjusted to remain conserved. A minimal choice is (E.1). However, this approach leaves a lot of ambiguity because the KdV charges could be mixed under this deformation. We take a different, more direct, approach here to show (E.1) that avoids this mixing ambiguity. As in (B.4), we will use the notation

$$[A(x), B(y)] = \sum_{n \geq 0} \mathcal{O}_n(A, B; y) \partial_x^n \delta(x-y). \tag{E.2}$$

Let us start with the left-hand side of (E.1). The action of a local spatial translation $y \mapsto y' = y + \epsilon(y)$ on a local operator $B$ is to shift it as

$$B(y') = B(y) + \left[ \int dx \, \epsilon(x) T_{xt}(x), B(y) \right] + O(\epsilon^2). \tag{E.3}$$

Integrating with measure $dy' = (1 + \partial_y \epsilon) dy$ gives

$$\int dy' \, B(y') = \int dy \left( B(y) + \partial_y \epsilon B(y) + \left[ \int dx \, \epsilon(x) T_{xt}(x), B(y) \right] \right) + O(\epsilon^2). \tag{E.4}$$

---

[19]The corrections to the currents coming from $Y_{\pm 1}$ in (D.6) were given in an explicit form in [4], while the terms coming from the $\mathcal{O}_n(x)$ were referred to as contact term corrections in a footnote.

To rescale space $L \to (1 + \varepsilon)L$ we take $\epsilon(x) = \varepsilon x$. We compute the commutator using (E.2):

$$
\begin{aligned}
\left[\int dx\, x\, T_{xt}(x), B(y)\right] &= \int dx\, x \sum_{n\geq 0} \mathcal{O}_n(T_{xt}, B; y) \partial_x^n \delta(x - y) \\
&= y\, \mathcal{O}_0(T_{xt}, B; y) - \mathcal{O}_1(T_{xt}, B; y) \\
&= i y [P, B(y)] - \mathcal{O}_1(T_{xt}, B; y).
\end{aligned}
\tag{E.5}
$$

Altogether,

$$
L\partial_L \int dy\, B(y) = \int dy\, \Big(B(y) - \mathcal{O}_1(T_{xt}, B; y)\Big) + [P, \mathcal{W}],
\tag{E.6}
$$

for $\mathcal{W} = i \int dy\, y\, B(y)$. In particular, taking $B = \frac{1}{2\pi}(T_{s+1} + \Theta_{s-1})$, whose integral is $P_s$, we get

$$
L\partial_L P_s = P_s - \frac{1}{2\pi} \int dy\, \mathcal{O}_1(T_{xt}, T_{s+1} + \Theta_{s-1}; y) + [P, \mathcal{W}].
\tag{E.7}
$$

Next we work out the right-hand side of (E.1). We compute

$$
\begin{aligned}
\partial_x(A_s^1 - A_s^{-1}) &\overset{(2.3)}{=} i[P_1 - P_{-1}, A_s^1 - A_s^{-1}] \overset{(2.4)}{=} i[P_s, T + \Theta - \bar{\Theta} - \bar{T}] = -2\pi[P_s, T_{xt}] \\
&\overset{(2.2)}{=} \int dy\, [T_{xt}(x), T_{s+1}(y) + \Theta_{s-1}(y)] \overset{(E.2)}{=} \sum_{n\geq 0} \partial_x^n \mathcal{O}_n(T_{xt}, T_{s+1} + \Theta_{s-1}; x).
\end{aligned}
\tag{E.8}
$$

The term $n = 0$ is a derivative, like the other terms:

$$
\mathcal{O}_0(T_{xt}, T_{s+1} + \Theta_{s-1}; x) = \int dy\, [T_{xt}(y), T_{s+1}(x) + \Theta_{s-1}(x)] = -\partial_x\big(T_{s+1}(x) + \Theta_{s-1}(x)\big),
\tag{E.9}
$$

so we get

$$
A_s^1 - A_s^{-1} = -\big(T_{s+1}(x) + \Theta_{s-1}(x)\big) + \sum_{n\geq 1} \partial_x^{n-1} \mathcal{O}_n(T_{xt}, T_{s+1} + \Theta_{s-1}; x),
\tag{E.10}
$$

up to shifts by multiples of the identity (the only local operator whose $\partial_x$ derivative vanishes). Then

$$
\frac{-1}{2\pi} \int dx \big(A_s^1 - A_s^{-1}\big) = P_s - \frac{1}{2\pi} \int dx\, \mathcal{O}_1(T_{xt}, T_{s+1} + \Theta_{s-1}; x).
\tag{E.11}
$$

We are done showing (E.1), because the right-hand sides of (E.7) and (E.11) agree up to $[P, \mathcal{W}]$.

# F  A comment on an integrability result

We show here that the evolution equation found in [58] using integrability describes some deformation that is outside the class of operator deformations that we study. Our results cannot be compared. Let us copy their equation for the $u$-th deformation here in our notations:

$$
\begin{aligned}
\partial_\lambda \langle P_k \rangle_n &= \pi^2 \left(L' \partial_L \langle P_k \rangle_n - k\, \theta_0' \langle P_k \rangle_n\right), \\
L' &\equiv \langle P_u \rangle_n + \langle P_{-u} \rangle_n - \pi^2(u - 1)\lambda\big(\langle P_u \rangle_n - \langle P_{-u} \rangle_n\big)\theta_0', \\
\theta_0' &\equiv -\frac{\langle P_u \rangle_n - \langle P_{-u} \rangle_n}{L - \pi^2(u - 1)\lambda\big(\langle P_u \rangle_n + \langle P_{-u} \rangle_n\big)}.
\end{aligned}
\tag{F.1}
$$

where we used the translation $I_u \to -P_u$, $\tau \to -\pi^2\lambda$, $R \to L$ and kept their notation for $L'$, $\theta_0'$. Because it reproduces results on the $T\bar{T}$ and $J\bar{T}$ deformations[20] the authors naturally suggested that for general spin $u$ it might describe the $X^{1,-u}$ (plus $X^{u,-1}$) deformations.

We give a general argument based on translation invariance that shows that (F.1) cannot correspond to adding to the action the integral $\partial_\lambda S = \int d^2x\, \mathcal{O}(x)$ of *any* local operator $\mathcal{O}$ and working with charges of local conserved currents. We then give a more restricted argument that the equation cannot describe $X^{u,-1}$ and/or $X^{1,-u}$ deformations, based on the observation that (F.1) does not involve the $A_k^{\pm u}$ operators. This might help determine what the deformation described by (F.1) actually is in the operator language.

## F.1 Nonlocality of the deformation or the charges

Assume that (F.1) described adding to the action the integral $\partial_\lambda S = \int d^2x\, \mathcal{O}(x)$ of a local operator $\mathcal{O}$ and working with charges of local conserved currents. Then invariance under translation along the (compact) spatial direction would be preserved, so momentum $P$ would remain quantized, hence $\lambda$ independent:

$$\langle P \rangle_n = \langle P \rangle_n^\circ, \tag{F.2}$$

by which we mean the momentum of the original CFT state $|n\rangle^\circ$. In the CFT, $P = -P_1 + P_{-1}$.

While in our framework we kept $-P_1 + P_{-1}$ equal to momentum $P$ (the quantized charge of spatial translation), (F.1) leads to

$$\langle -P_1 + P_{-1} \rangle_n \overset{(\text{F.1})}{=} \langle P \rangle_n^\circ - \frac{2\pi^2\lambda}{L}\langle P_u P_{-1} - P_1 P_{-u} \rangle_n + O(\lambda^2), \tag{F.3}$$

where we simplified a derivative by using that momentum depends on $L$ as $\langle -P_1 + P_{-1} \rangle_n^\circ \sim 1/L$. In an updated version of [58] another momentum $\check{P}$ is also defined, and it is found not to depend on $\lambda$ and hence coincides with the momentum we are using in the main text. The relation between $\check{P}$ and $P$ in [58] is the same (to linear order) as what we find in (F.3); what we are showing below is that $\check{P}$ and $P$ defined in [58] cannot both be integrals of local currents.

We would thus have two conserved charges: $-P_1 + P_{-1}$, and momentum $P$. Their difference would be a conserved charge as well, namely there would exist a conserved current $J_\mu$ such that (we divided by $\pi\lambda$ for later convenience)

$$\langle J_t \rangle_n = \frac{4\pi^2}{L^2}\langle P_u P_{-1} - P_1 P_{-u} \rangle_n + O(\lambda). \tag{F.4}$$

Notice in passing that for $u = 1$ the right-hand side cancels out and one can simply have $J_\mu = 0$. For $u \neq 1$ there is no cancellation and the right-hand side is the eigenvalue of $P_u P_{-1} - P_1 P_{-u}$ in the state $|n\rangle$. Each $P_k$ is an integral of a local operator over the spatial circle, so this quadratic combination is an integrated two-point function of components of currents. There is no reason to expect such an integrated two-point function to reduce to the one-point function of a well-chosen operator.

Let us make the argument sharp when starting from a CFT, for instance a minimal model: after all, the integrability results apply equally well to these theories. In a CFT with no further symmetry the KdV charges have odd spins $u \in 2\mathbb{Z} + 1$.

Consider first $u > 0$ and focus on a primary state with conformal dimensions $h, \bar{h}$. In that state, $\langle P_u \rangle_n^\circ = (2\pi/L)^u((-h)^{(u+1)/2} + \dots)$ and $\langle P_{-u} \rangle_n^\circ = (2\pi/L)^u((-\bar{h})^{(u+1)/2} + \dots)$ are polynomials of degree $(u+1)/2$ in $h$ and $\bar{h}$, respectively, so

$$\langle J_t \rangle_n = (-1)^{(u+3)/2}\Big(\frac{2\pi}{L}\Big)^{u+3}\big(\bar{h}h^{(u+1)/2} - h\bar{h}^{(u+1)/2} + \dots\big) + O(\lambda). \tag{F.5}$$

---

[20]More precisely, for a CFT the $u \to 0$ limit has a four-parameter generalization, and a choice of these parameters gives the usual $J\bar{T}$ deformation.

In a generic CFT, conserved charges split into a sum of a holomorphic and an antiholomorphic charges, and their one-point function in a primary state is of the form $f(h,c) + g(\bar{h},c)$. For $u > 1$, (F.5) is not of this form, so the current $J_\mu$ cannot exist. This concludes our proof in that case.

For $u < 0$, the matrix element $\langle J_t \rangle_n$ is a sum of terms $\langle P_u P_{-1} \rangle_n^\circ$ and $\langle P_1 P_{-u} \rangle_n^\circ$ that each involve only one of the chiral Virasoro algebras. However there is no way to write these terms as the expectation value of a local conserved current. Let us see this explicitly for $u = -1$. Note that since $|n\rangle$ is an eigenstate of $P_1$,

$$\langle P_1 \rangle_n^2 = \langle P_1^2 \rangle_n = \langle (L_0 - c/24)^2 \rangle_n \quad \text{at } \lambda = 0 \,, \tag{F.6}$$

which cannot be equal for all states to a linear combination of

$$\int dx \, {:}\partial^k T \partial^l T{:} = \# L_0^2 + \# L_0 + \# + 2i^{k-l} \sum_{m=1}^{\infty} m^{k+l} L_{-m} L_m \tag{F.7}$$

because the sum of $L_{-m} L_m$ cannot cancel in all states.

We conclude that (F.1) cannot describe in general for $u \neq 1$ the evolution of local charges under a deformation that respects periodic translation invariance and locality. If (F.1) describes the effect of field-dependent changes of coordinates as proposed in [58], then it is perhaps not surprising that periodicity of the space coordinate is not preserved. It may be the case that the deformation only makes sense on the plane rather than the cylinder. Another possibility may be that the charges $\langle P_k \rangle_n$ appearing in (F.1) are not integrals of *local* conserved currents.

## F.2 Linear order around a CFT

While our proof above rules out deformations by arbitrary local operators it is instructive to look more carefully at why the integrability equation (F.1) does not correspond to a deformation by $X$ operators.

Let us consider deformations of CFT to linear order by a combination of $T_{u+1}\bar{T}$ and $T\bar{T}_{u+1}$ (for $u > 0$), namely by $\alpha X^{-1,u} + \beta X^{1,-u}$ for some coefficients $\alpha, \beta$. As we explained, our formalism expresses the variation of KdV charges in terms of operators $A_s^t$. In a CFT, these operators vanish when signs of $s$ and $t$ differ, and furthermore they have the symmetry $t A_s^t = s A_t^s$ derived in (B.3). This allows us to write (3.1) as

$$\langle P_k \rangle_n = \langle P_k \rangle_n^\circ + \pi \lambda \Big( \frac{-2\pi \alpha k}{L} \langle P_u \rangle_n^\circ \langle P_k \rangle_n^\circ + \beta \langle P_1 \rangle_n^\circ \langle A_k^{-u} \rangle_n^\circ \Big) + O(\lambda^2) \qquad \text{for } k < 0 \,,$$

$$\langle P_k \rangle_n = \langle P_k \rangle_n^\circ + \pi \lambda \Big( -\frac{2\pi \beta k}{L} \langle P_{-u} \rangle_n^\circ \langle P_k \rangle_n^\circ + \alpha \langle P_{-1} \rangle_n^\circ \langle A_k^{u} \rangle_n^\circ \Big) + O(\lambda^2) \qquad \text{for } k > 0 \,, \tag{F.8}$$

where the superscript $\circ$ denotes CFT quantities. At this point we must remember that (3.1) is only one choice of how to deform KdV charges in such a way as to keep them conserved: one can add to it other conserved charges of the CFT, as discussed in detail in Appendix C.[21]

In contrast, using that the $k$-th KdV charge scales as $L^{-|k|}$ in the CFT, the integrability result (F.1) gives

$$\langle P_k \rangle_n \overset{\text{(F.1)}}{=} \langle P_k \rangle_n^\circ - \frac{2\pi^2 \lambda k}{L} \langle P_u \rangle_n^\circ \langle P_k \rangle_n^\circ + O(\lambda^2), \qquad \text{for } k < 0 \,,$$

$$\langle P_k \rangle_n \overset{\text{(F.1)}}{=} \langle P_k \rangle_n^\circ + \frac{2\pi^2 \lambda k}{L} \langle P_{-u} \rangle_n^\circ \langle P_k \rangle_n^\circ + O(\lambda^2), \qquad \text{for } k > 0 \,. \tag{F.9}$$

---

[21] In fact, dimensional analysis and spurion analysis together rule out such mixing for the $\alpha X^{-1,u} + \beta X^{1,-u}$ deformation ($u > 0$). Since $X^{s,t}$ ($s, t > 0$) vanish in a CFT, it is not possible to distinguish (at linear order around the CFT) the $\alpha X^{-1,u} + \beta X^{1,-u}$ deformation from a sum of this deformation and of any $X^{s,t}$ ($s, t > 0$). While the couplings of $X^{s,t}$ are invisible in the Hamiltonian at this order, they weaken dimensional and spurion analysis because of their varied dimensions and spins. These couplings allow a large class of mixing ambiguities. We thus move on with the proof without using dimensional and spurion analysis.

In both of these lines we recognize one of the terms in (F.8) (with $\alpha = 1 = -\beta$) but not the term $\langle P_{-1} \rangle_n^\circ \langle A_k^u \rangle_n^\circ$ for $k > 0$ (and its complex conjugate for $k < 0$). As discussed in Section 4.2, $\langle A_k^u \rangle_n^\circ$ cannot be determined from the integrals of motion $\langle P_k \rangle_n^\circ$. What is less immediate is whether the term $\langle P_{-1} \rangle_n^\circ \langle A_k^u \rangle_n^\circ$ could be fully absorbed by the freedom to shift $\partial_\lambda \langle P_k \rangle_n$ by a conserved charge,[22] possibly combined with a change of $\alpha, \beta$.

This can be ruled out tediously in an ad-hoc manner by considering the case where $|n\rangle$ is a primary state of conformal dimensions $h, \bar{h}$ and working out the leading powers of $h$ and $\bar{h}$ in each expectation value. The question then boils down to whether there could be some coefficient $\gamma$ such that

$$
\begin{aligned}
&\langle P_{-1} \rangle_n^\circ \langle A_k^u \rangle_n^\circ + \gamma \langle P_{-u} \rangle_n^\circ \langle P_k \rangle_n^\circ \\
&= \#\big(\bar{h} + \dots\big)\big(h^{(u+k)/2} + \dots\big) + \#\gamma\big(\bar{h}^{(u+1)/2} + \dots\big)\big(h^{(k+1)/2} + \dots\big)
\end{aligned}
\tag{F.10}
$$

is $\langle Q \rangle_n^\circ$ for some conserved charge $Q$ (here $\#$ denote known coefficients). Since the leading monomials cannot cancel for $u > 1$, by the same logic as around (F.5), (F.10) does not have the form $f(h, c) + g(\bar{h}, c)$ of the expectation value of a conserved charge.

Thus, (F.1) would need significant modifications involving $\langle A_k^u \rangle_n$ to describe the $T_{u+1}\bar{T}$ or $T\bar{T}_{u+1}$ deformations.

# G  Nonabelian symmetries

In the main text we exclusively work with a chosen commuting subset of the conserved charges. Here we discuss what changes for charges $Q_a$ that do not commute. Most prominenly this includes non-abelian flavor symmetries. Another example is the full set of monomials built from $T$ and its derivative in a CFT: this forms a non-abelian extension of the KdV charges.

We learn that it only makes sense to deform by bilinears combinations $X_{ab}$ of currents when the corresponding charges $Q_a$ and $Q_b$ commute. Along the deformation, one can preserve the charges $Q_c$ that commute with both of these, and the structure constants of these charges are not deformed. For instance, the $T\bar{T}$ deformation preserves the full charge algebra (non-abelian flavor symmetries and perhaps more surprisingly the non-abelian KdV charge algebra of a CFT) including its structure constants.

## G.1  The operators $A$

We denote structure constants as $f_{ab}{}^c$, so that $[Q_a, Q_b] = f_{ab}{}^c Q_c$.

Because $[Q_a, Q_b] - f_{ab}{}^c Q_c = 0$, the integral of $[Q_a, J_{b,z}dz - J_{b,\bar{z}}d\bar{z}] - f_{ab}{}^c(J_{c,z}dz - J_{c,\bar{z}}d\bar{z})$ on any cycle vanishes, hence this one-form is exact. Namely,

$$
\begin{aligned}
&[Q_a, J_{b,z}(z, \bar{z})] - f_{ab}{}^c J_{c,z}(z, \bar{z}) = -i\partial A_{ab}(z, \bar{z}), \\
&[Q_a, J_{b,\bar{z}}(z, \bar{z})] - f_{ab}{}^c J_{c,\bar{z}}(z, \bar{z}) = i\bar{\partial} A_{ab}(z, \bar{z}),
\end{aligned}
\tag{G.1}
$$

where $A_{ab}$ are some (local) operators defined up to shifts by multiples of the identity. We also denote $A_{ab} = A(Q_a, J_b)$ to emphasize that the operator depends on a choice of charge and a choice of current, which are two somewhat asymmetric inputs. The $A_s^t$ operators considered in the main text are special cases of $A_{ab}$. With this notation it is easy to check that

$$
A(P_1, J) = J_z \quad \text{and} \quad A(P_{-1}, J) = J_{\bar{z}},
\tag{G.2}
$$

---

[22]In fact, this essentially happens in Section 4. To linear order around a CFT the deformation studied there is $X^{-1,u}$, corresponding to $\alpha = 1$ and $\beta = 0$ here, and we focus there on the zero-momentum sector. In that sector we can check $\langle P_{-1} \rangle_n^\circ \langle A_k^u \rangle_n^\circ = \langle P_1 \rangle_n^\circ \langle A_k^u \rangle_n^\circ = (L/2\pi)\langle \mathcal{X}_{1k}^{1u} \rangle_n^\circ + \langle P_k \rangle_n^\circ \langle P_u \rangle_n^\circ (2\pi/L)$. The first term is a shift by the conserved charge of the holomorphic current $\mathcal{X}_{1k}^{1u}$. The second is expressed in terms of charges that we have control on. Away from the zero-momentum sector this switch to holomorphic quantities is not possible.

up to the shift-by-identity freedom. Improving the currents affects $A(Q_a, J_b)$ as follows:

$$(J_{d,z}, J_{d,\bar{z}}) \to (J_{d,z} + \partial \mathcal{O}_d, J_{d,\bar{z}} - \bar{\partial} \mathcal{O}_d) \implies A(Q_a, J_b) \to A(Q_a, J_b) + i[Q_a, \mathcal{O}_b] - i f_{ab}{}^c \mathcal{O}_c . \text{ (G.3)}$$

In another appendix we showed a symmetry property (A.4) $[P_{[s}, A^u_{t]}] = 0$ for the case of commuting charges. To show it the main point was to show the $\partial$ and $\bar{\partial}$ derivatives vanished. Let us follow the same strategy when structure constants are non-zero. We work out

$$\begin{aligned}
-2i\partial[Q_{[a}, A_{b]c}] &= 2[Q_{[a}, [Q_{b]}, J_{c,z}]] - 2[Q_{[a}, f_{b]c}{}^d J_{d,z}] \\
&= f_{ab}{}^d [Q_d, J_{c,z}] - f_{bc}{}^d [Q_a, J_{d,z}] + f_{ac}{}^d [Q_b, J_{d,z}] \\
&= -i\partial(f_{ab}{}^d A_{dc} - f_{bc}{}^d A_{ad} + f_{ac}{}^d A_{bd}),
\end{aligned} \text{ (G.4)}$$

where the first equality is the definition (G.1), the second equality uses the Jacobi identity and $[Q_a, Q_b] = f_{ab}{}^d Q_d$, and the last equality expresses each commutator $[Q_a, J_{b,z}] = f_{ab}{}^c J_{c,z} - i\partial A_{ab}$ before using a cancellation $f_{ab}{}^d f_{dc}{}^e - f_{bc}{}^d f_{ad}{}^e + f_{ac}{}^d f_{bd}{}^e = 0$ that is due to the Jacobi identity $[[Q_a, Q_b], Q_c] - [Q_a, [Q_b, Q_c]] + [Q_b, [Q_a, Q_c]] = 0$. Together with the analogous result for $i\bar{\partial}$, this means that

$$[Q_a, A_{bc}] - [Q_b, A_{ac}] - (f_{ab}{}^d A_{dc} + f_{cb}{}^d A_{ad} + f_{ac}{}^d A_{bd}) \text{ (G.5)}$$

is a translationally-invariant but local operator, hence a multiple of the identity. This reduces to the definition of $A_{bc}$ upon specializing to $Q_a \to P_{\pm 1}$ and using (G.2): this uses that structure constants $f_{ab}{}^c$ vanish when $Q_a = P_{\pm 1}$, because any conserved charge commutes by definition with these charges.[23] Another interesting case is when $Q^a$, $Q^b$ and $Q^c$ commute. Then all structure constants drop out, so the operator is traceless[24] hence vanishes. In other words,

$$[Q, A(Q', J'')] = [Q', A(Q, J'')] \quad \text{when } [Q, Q'] = [Q, Q''] = [Q', Q''] = 0 . \text{ (G.6)}$$

## G.2 The operators $X$ and deformations

Consider a pair of conserved currents $J_a$ and $J_b$. For the same reason as the usual $T\bar{T} - \Theta\bar{\Theta}$ collision, we can define $X_{ab} = (\epsilon^{\mu\nu} J_{a,\mu} J_{b,\nu})_{\text{reg}}$ by point-splitting, modulo total derivatives. Indeed, conservation leads to

$$\begin{aligned}
&\partial_{\bar{z}}(J_{a,z}(z,\bar{z}) J_{b,\bar{w}}(w,\bar{w}) - J_{a,\bar{z}}(z,\bar{z}) J_{b,w}(w,\bar{w})) \\
&= (\partial_z + \partial_w)(J_{a,z}(z,\bar{z}) J_{b,\bar{w}}(w,\bar{w})) + (\bar{\partial}_{\bar{z}} + \bar{\partial}_{\bar{w}})(J_{a,\bar{z}}(z,\bar{z}) J_{b,w}(w,\bar{w})),
\end{aligned} \text{ (G.7)}$$

hence the collision $\epsilon^{\mu\nu} J_{a,\mu} J_{b,\nu}$ is independent of the offset $(z-w, \bar{z}-\bar{w})$, modulo total derivatives. Amusingly we did not need to assume that the charges $Q_a$ and $Q_b$ commute.

Now deform the action by $X_{ab}$. The key question is which symmetries $Q_c$ can be preserved. As we showed in (2.12), the condition is that $[Q_c, X_{ab}]$ needs to be a total derivative. One can compute

$$[Q_c, X_{ab}] = f_{ca}{}^d X_{db} + f_{cb}{}^d X_{ad} + i\partial(J_{a,\bar{z}} A_{cb} - A_{ca} J_{b,\bar{z}})_{\text{reg}} + i\bar{\partial}(J_{a,z} A_{cb} - A_{ca} J_{b,z})_{\text{reg}} . \text{ (G.8)}$$

A word of warning: the bilinears $J_{a,\mu} A_{cb} - A_{ca} J_{b,\mu}$ regulated by point splitting have significantly more ambiguities than those we discuss in Appendix A.2 for the case of commuting charges.

In order for the deformation to make sense beyond linear order, the symmetries $Q_a$ and $Q_b$ that define the deformation must themselves be preserved by the deformation. Setting

---

[23]To be more precise this assumes that currents do not depend explicitly on coordinates; otherwise the conservation equation $\partial_t Q_a = 0$ and the trivial equation $\partial_x Q_a = 0$ do not translate to $[P_{\pm 1}, Q_a] = 0$.

[24]In this infinite-dimensional setting the trace is ill-defined. One can consider instead the expectation value in any common eigenstate of $Q^a$ and $Q^b$.

$c = a$ and $c = b$ we see that the above commutator is only a total derivative if $[Q_a, Q_b]$ is both proportional to $Q_a$ and to $Q_b$, hence is simply zero.

We learn that it only makes sense to deform by bilinears $X_{ab}$ of commuting currents.

Then, apart from fine-tuned cases where $f_{ca}{}^d X_{db} + f_{cb}{}^d X_{ad}$ somehow cancels, the charges that are preserved by the $X_{ab}$ deformation are the charges $Q_c$ that commute with $Q_a$ and $Q_b$. An important special case is for the $T\bar{T}$ deformation: charges can be preserved if and only if they commute with $P_{\pm 1}$, namely the corresponding currents do not depend explicitly on coordinates.

### G.3 Structure constants are preserved

Under a deformation by $X_{ab}$ (with $[Q_a, Q_b] = 0$), consider two charges $Q_c$ and $Q_d$ that commute with $Q_a$ and $Q_b$. In other words, these four charges commute pairwise except $Q_c$ and $Q_d$, whose commutator we wish to study. Ignoring regulator terms (which work out in the same way as explained in Appendix A.2) we have

$$
\begin{aligned}
\delta[Q_c, Q_d] &= [Q_c, \delta Q_d] + [\delta Q_c, Q_d] \\
&= \frac{i}{2} \int dx \left[ Q_c, J_{a,t} A_{db} - A_{da} J_{b,t} \right] - (c \leftrightarrow d) \\
&= \frac{i}{2} \int dx \left( [Q_c, J_{a,t}] A_{db} + J_{a,t}[Q_c, A_{db}] - [Q_c, A_{da}] J_{b,t} - A_{da}[Q_c, J_{b,t}] - \right. \\
&\qquad\qquad \left. - (c \leftrightarrow d) \right),
\end{aligned}
\tag{G.9}
$$

where we simply expanded the commutators. Rewriting the commutators $[Q, J'_t] = \partial_x A(Q, J')$, and using (G.5) to rewrite $[Q_c, A_{db}] - [Q_d, A_{cb}] = f_{cd}{}^e A_{eb}$ (other structure constants vanish), we get

$$
\delta[Q_c, Q_d] = \frac{i}{2} \int dx \left( \partial_x A_{ca} A_{db} - \partial_x A_{da} A_{cb} + J_{a,t} f_{cd}{}^e A_{eb} - f_{cd}{}^e A_{ea} J_{b,t} - A_{da} \partial_x A_{cb} + A_{ca} \partial_x A_{db} \right).
\tag{G.10}
$$

The first two and last two terms combine into $x$ derivatives, while the middle two terms are simply $f_{cd}{}^e \delta Q_e$. Altogether, $\delta\left( [Q_c, Q_d] - f_{cd}{}^e Q_e \right) = 0$, namely structure constants do not change. This is in harmony with the conjecture in [4] that the $T\bar{T}$ deformation leaves the KdV charges commuting, which we showed in (2.17) in a less abstract language.

In Appendix C we analyze ambiguities that affect the definition of currents, charges and $A_{ab}$ appearing throughout the paper. In this appendix we worked with the specific fixing of ambiguities and saw that the symmetry algebra remains undeformed. If we were to reintroduce ambiguities, the nonabelian structure would get deformed. Hence, if a nonabelian algebra is preserved, requiring it to remain undeformed is an efficient principle to fix the ambiguities.

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
