# Peer review of "KdV charges in TTbar theories and new models with super-Hagedorn behavior"

_SciPost Physics, doi:SciPost Phys. 7, 043 (2019)_

## Round 1 · Referee Report · Anonymous (Referee 1) · 2019-7-27

Report
This paper represents a useful and important addition to the literature on TTbar. In particular it addresses some open questions in the earlier paper of Smirnov and Zamolodchikov about the fate of the higher spin 'KdV' conserved charges in a CFT. The authors find a beautiful result that they are slave to the TTbar flow in the same way that a passive scalar follows the Burgers flow. They also discuss other related deformations and the non-Lorentz invariant case.
The paper is very thorough if rather dense, and I have no suggestions for improvements.
The paper is very thorough if rather dense, and I have no suggestions for improvements.

Author: Bruno Le Floch on 2019-09-26 [id 609]
(in reply to Report 3 on 2019-09-12)Thank you for the report. In the proofs we clarified our reference to [5,57] by a footnote, in which we explain that having infinitely many conserved charges does not make the theory integrable in the sense that dynamics of the theory would be completely fixed by symmetries. For instance it is expected that a generic CFT is non-integrable, despite the scarcity of explicit examples.

---

## Round 1 · Referee Report · Anonymous (Referee 3) · 2019-9-12

Report
The manuscript contains two main results: the operator-language derivation of the formula for the expectation values of KdV charges in $T\bar T$ deformed theories and the derivation of the zero-momentum energy spectrum in theories deformed by certain bilinears of KdV currents. I agree with the previous referees that this paper should be published in its present form. It contains several technically challenging calculations and the authors have developed some interesting novel techniques to perform them.
The only thing that made me slightly uncomfortable is the way the authors refer to [5,57]. It is true that there the evolution equation for KdV charges was performed using integrability, however, in this case it does not make the derivation less general. The reason is that, under rather mild assumptions, existence of a single higher spin charge leads to existence of infinitely many and hence integrability. Thus results of [5,57] couldn't have been a "miracle of some special models". I do not think that this fact reduces the value of the present paper in any way. Direct operator-language derivation presented here is of its own interest, Independently of the previous results that rely on heavy integrability machinery.
The only thing that made me slightly uncomfortable is the way the authors refer to [5,57]. It is true that there the evolution equation for KdV charges was performed using integrability, however, in this case it does not make the derivation less general. The reason is that, under rather mild assumptions, existence of a single higher spin charge leads to existence of infinitely many and hence integrability. Thus results of [5,57] couldn't have been a "miracle of some special models". I do not think that this fact reduces the value of the present paper in any way. Direct operator-language derivation presented here is of its own interest, Independently of the previous results that rely on heavy integrability machinery.
Requested changes
Authors may wish to change how they refer to [5,57].
Report
This is a highly professional, well-written paper. The results on the super-Hagedorn behaviour of the integrable perturbations which break the Lorentz invariance are original and interesting. I do recommend this paper for publication.

---

## Editorial Decision

published